# Is Score Matching Suitable for Estimating Point Processes?

**Haoqun Cao**[1], **Zizhuo Meng**[2], **Tianjun Ke**[1], **Feng Zhou**[1,3]*
[1]Center for Applied Statistics and School of Statistics, Renmin University of China
[2]Data Science Institute, University of Technology Sydney
[3]Beijing Advanced Innovation Center for Future Blockchain and Privacy Computing
`hcao65@wisc.edu, feng.zhou@ruc.edu.cn`

## Abstract

Score matching estimators have gained widespread attention in recent years partly because they are free from calculating the integral of normalizing constant, thereby addressing the computational challenges in maximum likelihood estimation (MLE). Some existing works have proposed score matching estimators for point processes. However, this work demonstrates that the incompleteness of the estimators proposed in those works renders them applicable only to specific problems, and they fail for more general point processes. To address this issue, this work introduces the weighted score matching estimator to point processes. Theoretically, we prove the consistency of our estimator and establish its rate of convergence. Experimental results indicate that our estimator accurately estimates model parameters on synthetic data and yields results consistent with MLE on real data. In contrast, existing score matching estimators fail to perform effectively. Codes are publicly available at `https://github.com/KenCao2007/WSM_TPP`.

## 1 Introduction

Point processes are a class of statistical models used to characterize event occurrences. Typical models include Poisson processes [9] and Hawkes processes [3]. Their applications span various fields such as seismology [16; 17], finance [1; 4], criminology [15], and neuroscience [11; 26]. In the field of point processes, maximum likelihood estimation (MLE) has been a conventional estimator. However, MLE has an inherent limitation: it requires the computation of the normalizing constant, which corresponds to the intensity integral term in the likelihood. Except for simple cases, calculating the intensity integral analytically is generally infeasible. This necessitates the use of numerical integration methods like Monte Carlo or quadrature for approximating the computation. This introduces approximation errors, and more importantly, for high-dimensional problems, numerical integration encounters the curse of dimensionality, rendering training infeasible.

To address this issue, prior research has introduced the concept of score matching (SM) [5] to the field of point processes. For instance, [18] derived the application of SM to the estimation of traditional statistical Poisson processes. Furthermore, [24] extended the use of SM to the estimation of deep covariate spatio-temporal point processes. [10] also generalized the application of SM to Hawkes processes. These works have greatly advanced the utilization of SM for point processes. However, in practical applications, we have found that these estimators only work for specific point processes. For more general cases, these estimators cannot accurately estimate model parameters, even for some simple statistical point processes. One of the core contribution of this work is to *theoretically* demonstrate the incompleteness of the estimators proposed in the aforementioned studies.

---

*Corresponding author.

38th Conference on Neural Information Processing Systems (NeurIPS 2024).

The incompleteness of the estimators in the aforementioned studies stems from the transition from explicit SM to implicit SM. The explicit SM estimates model parameters by minimizing the expected distance between the gradient of the log-density of the model and the gradient of the log-density of the data. However, we cannot directly minimize the above objective function since it depends on the unknown data distribution. To facilitate solving, we need to convert the above explicit SM to implicit SM by using a trick of integration by parts, provided that some regularity conditions are satisfied [5]. In [18; 24; 10], they assume that the required regularity conditions are satisfied in their point process models and directly employ the implicit SM objective. However, as demonstrated in Section 3, the required regularity conditions cannot be met for general point processes. This implies that the concise implicit SM objectives (Equation (2) in [18], Equation (10) in [24], Equation (4) in [10]) are incomplete, and they cannot accurately estimate parameters for general point processes.

To address this issue, this work introduces a (autoregressive) weighted score matching (WSM) estimator that can be applied to more general point processes. WSM eliminates the intractable terms in SM objective by adding a weight function that takes zero at the boundary of the integration region. Compared to previous work on WSM [6; 22; 12], we are the first work to apply WSM on a stochastic process where the dimension $N_T$ is also random. This stochasticity in dimensionality poses greater challenges to the derivation, requiring special treatment to address this issue.

Specifically, we make following contributions: **(1)** We theoretically demonstrate that implicit (autoregressive) SM estimators in [18; 24; 10] are incomplete because the required regularity conditions cannot be satisfied for general point processes. **(2)** To address this issue, we propose a (autoregressive) WSM estimator that is applicable to general point processes. Theoretically, we establish its consistency and convergence rate. **(3)** In experiments, we confirm that on synthetic data, (autoregressive) WSM successfully recovers the ground-truth parameters; on real data, (autoregressive) WSM estimates results consistent with MLE; while existing (autoregressive) SM estimator fails in both scenarios.

## 2 Preliminaries

Now we provide knowledge on Poisson and Hawkes processes and (autoregressive) score matching.

### 2.1 Poisson Process and Hawkes Process

The Poisson process [9] is a stochastic point process that models the occurrence of events over a time window $[0, T]$. A trajectory from Poisson process can be represented as an ordered sequence $\mathcal{T} = (t_1, \ldots, t_{N_T})$ where $N_t = \max\{n : t_n \leq t, t \in [0, T]\}$ is the corresponding counting process and thus $N_T$ is the random number of events in $[0, T]$. The inhomogeneous Poisson process has a time-varying intensity $\lambda(t)$ representing the instantaneous rate of event occurrence at $t$. Mathematically, the intensity function is defined as $\lambda(t) = \lim_{\delta_t \to 0} \mathbb{E}[N_{t+\delta_t} - N_t]/\delta_t$. The probability density function of Poisson process is:

$$p(\mathcal{T}) = \prod_{n=1}^{N_T} \lambda(t_n) \exp\left(-\int_0^T \lambda(t)dt\right). \tag{1}$$

The Hawkes process [3] is a self-excitation point process where the occurrence of an event increases the likelihood of more events in the future. A trajectory from Hawkes process is similarly represented as $\mathcal{T} = (t_1, \ldots, t_{N_T})$ on $[0, T]$. The conditional intensity function of Hawkes process, representing the instantaneous rate of event occurrence at $t$ given the history up to but not including $t$, is:

$$\lambda^*(t) = \lambda(t|\mathcal{F}_{t^-}) = \mu(t) + \sum_{t_j < t} g(t - t_j), \tag{2}$$

where $\mu(t)$ is the baseline intensity, $g(\cdot)$ is the triggering kernel representing the self-excitation effect, the summation expresses the accumulative excitation from all past events, $\mathcal{F}_{t^-}$ is the historical information up to but not including $t$. $\lambda^*(t)$ means the intensity is dependent on the history.

Poisson process assumes the independence of event occurrences, while Hawkes process extends it by introducing an autoregressive structure, making subsequent events dependent on prior events. Given

the history $\mathcal{F}_{t_n} = (t_1, \ldots, t_n)$, the conditional probability density of $(n+1)$-th event at $t > t_n$ is:

$$p(t|\mathcal{F}_{t_n}) = \lambda^*(t) \exp\left(-\int_{t_n}^t \lambda^*(\tau)d\tau\right). \tag{3}$$

Here, we introduce the definition of the univariate Hawkes process. However, multivariate Hawkes processes also exist. For ease of notation, we use the univariate case for illustration, but we also provide solutions for the multivariate case.

## 2.2 Score Matching

MLE is a classic estimator that minimizes Kullback–Leibler divergence between a model distribution and data distribution. However, a drawback is the intractable computation of the normalizing constant. Approximating it through numerical integration can be computationally demanding. In contrast, SM [5] offers an alternative by minimizing Fisher divergence between model and data distributions:

$$\mathcal{L}_{\text{SM}}(\theta) = \frac{1}{2}\mathbb{E}_{p(\mathbf{x})}\|\nabla_{\mathbf{x}} \log p(\mathbf{x}) - \nabla_{\mathbf{x}} \log p_\theta(\mathbf{x})\|^2,$$

where $p(\mathbf{x})$ represents the data distribution, $p_\theta(\mathbf{x})$ is the parameterized model distribution, the gradient of the log-density is called the score, and $\|\cdot\|$ represents a suitable norm, such as the $\ell^2$ norm. Minimizing the Fisher divergence above provides the parameter estimate. The advantage of SM lies in its ability to bypass the computation of the normalizing constant since the score no longer contains this constant: $p_\theta(\mathbf{x}) = \frac{1}{Z(\theta)}\tilde{p}_\theta(\mathbf{x})$ where $Z(\theta) = \int \tilde{p}_\theta(\mathbf{x})d\mathbf{x}$, $\nabla_{\mathbf{x}} \log p_\theta(\mathbf{x}) = \nabla_{\mathbf{x}} \log \tilde{p}_\theta(\mathbf{x})$.

Under certain conditions, we can use integration by parts to replace the explicit SM objective, which involves an unknown distribution $p(\mathbf{x})$, with an equivalent implicit one,

$$\mathcal{J}_{\text{SM}}(\theta) = \mathbb{E}_{p(\mathbf{x})}\left[\frac{1}{2}\|\nabla_{\mathbf{x}} \log p_\theta(\mathbf{x})\|^2 + \text{Tr}\left(\nabla_{\mathbf{x}}^2 \log p_\theta(\mathbf{x})\right)\right]. \tag{4}$$

## 2.3 Autoregressive Score Matching

An autoregressive model defines a probability density $p(\mathbf{x})$ as a product of conditionals using the chain rule: $p(\mathbf{x}) = \prod_{n=1}^N p(x_n|\mathbf{x}_{<n})$, where $x_n$ is the $n$-th entry and $\mathbf{x}_{<n}$ denotes the entries with indices smaller than $n$. The original SM is not suitable for autoregressive models because the autoregressive structure introduces challenges in gradient computation in Equation (4). To address this issue, [13] proposed autoregressive score matching (ASM). Unlike SM, which minimizes the Fisher divergence between the joint distributions of the model $p_\theta(\mathbf{x})$ and the data $p(\mathbf{x})$, ASM minimizes the Fisher divergence between the conditionals of the model $p_\theta(x_n|\mathbf{x}_{<n})$ and the data $p(x_n|\mathbf{x}_{<n})$:

$$\mathcal{L}_{\text{ASM}}(\theta) = \frac{1}{2}\sum_{n=1}^N \mathbb{E}_{p(\mathbf{x}_{\leq n})}\left(\frac{\partial \log p(x_n|\mathbf{x}_{<n})}{\partial x_n} - \frac{\partial \log p_\theta(x_n|\mathbf{x}_{<n})}{\partial x_n}\right)^2.$$

Similarly, the above explicit ASM objective involves an unknown distribution $p(x_n|\mathbf{x}_{<n})$. Under specific regularity conditions, we can apply integration by parts to derive an implicit ASM objective:

$$\mathcal{J}_{\text{ASM}}(\theta) = \sum_{n=1}^N \mathbb{E}_{p(\mathbf{x}_{\leq n})}\left[\frac{1}{2}\left(\frac{\partial \log p_\theta(x_n|\mathbf{x}_{<n})}{\partial x_n}\right)^2 + \frac{\partial^2 \log p_\theta(x_n|\mathbf{x}_{<n})}{\partial x_n^2}\right]. \tag{5}$$

# 3 Score Matching for Poisson Process

We analyze the application of SM for Poisson process and its failure in achieving consistent estimation. Subsequently, we propose a provably consistent WSM estimator.

## 3.1 Failure of Score Matching for Poisson Process

Consider a Poisson process $\mathcal{T} = (t_1, \ldots, t_{N_T})$ on $[0, T]$. Let $p(\mathcal{T})$ represent the data distribution, which is uniquely associated with an intensity function $\lambda(t)$. Let $p_\theta(\mathcal{T})$ represent the parameterized

model distribution, which is uniquely associated with a parameterized intensity function $\lambda_\theta(t)$. In the following, we denote the score as $\psi(t_n) = \frac{\partial}{\partial t_n} \log p(\mathcal{T})$.

Previous works [18; 24] have both attempted to apply SM to the Poisson process:

$$\mathcal{L}_{\text{SM}}(\theta) = \frac{1}{2} \mathbb{E}_{p(\mathcal{T})} \left[ \sum_{n=1}^{N_T} (\psi(t_n) - \psi_\theta(t_n))^2 \right]. \tag{6}$$

In order for SM to be practical, [18; 24] assumed that specific regularity conditions are satisfied. Therefore, they employed an implicit SM objective similar to Equation (4):

$$\mathcal{J}_{\text{SM}}(\theta) = \mathbb{E}_{p(\mathcal{T})} \left[ \sum_{n=1}^{N_T} \frac{1}{2} \psi_\theta^2(t_n) + \frac{\partial \psi(t_n)}{\partial t_n} \right]. \tag{7}$$

In practical applications, we have found that the above estimator works only for specific Poisson processes and fails for more general Poisson processes. The reason for its failure lies in the fact that, for more general Poisson processes, the specific regularity conditions cannot be satisfied.

Such conditions require the probability density function of the random variable is zero when it approaches infinity in any of its dimensions. However, for point processes, such requirement is not satisfied, because the random variable in point process $\mathcal{T} = (t_1, \dots, t_{N_T})$ is not of fixed dimension and takes values in a subset of $\mathbb{R}_+^{N_T}$. Therefore, for general Poisson processes, we cannot derive the implicit SM in Equation (7) based on the explicit SM in Equation (6).

**Proposition 3.1.** *Assume that all functions and expectations in $\mathcal{L}_{SM}(\theta)$ and $\mathcal{J}_{SM}(\theta)$ are well defined, we have,*

$$\begin{aligned}
\mathcal{L}_{SM}(\theta) = {} & \mathcal{J}_{SM}(\theta) + const - \sum_{N=1}^{\infty} \int p(t_1, \dots, t_N) \frac{\partial \log p_\theta(t_1, \dots, t_N)}{\partial t_1} \bigg|_{t_1=0} d\mathcal{T}_{2:N} \\
& + \sum_{N=1}^{\infty} \int p(t_1, \dots, t_N) \frac{\partial \log p_\theta(t_1, \dots, t_N)}{\partial t_N} \bigg|_{t_N=T} d\mathcal{T}_{1:N-1}.
\end{aligned} \tag{8}$$

*Therefore, $\mathcal{L}_{SM}(\theta)$ is equivalent to $\mathcal{J}_{SM}(\theta)$ if and only if the sum of the last two terms is a constant not containing $\theta$.*

For specific Poisson processes, the sum of the last two terms can be zero. However, for more general cases, this sum contains $\theta$. This implies that $\mathcal{J}_{\text{SM}}$ fails for general Poisson processes.

### 3.2 Weighted Score Matching

To address the situation where SM fails, inspired by [6; 22], we introduce the WSM for Poisson process. The core idea of WSM is to eliminate the two intractable terms by adding a weight function that takes zero at the boundary of the integration region. The weight function is designed to be a vector-valued function $\mathbf{h} : \mathbb{R}_+^{N_T} \to \mathbb{R}_+^{N_T}$ with the $n$-th element denoted as $h_n(\mathcal{T})$. Here, we present the conditions that a valid weight function should satisfy:

$$\lim_{t_n \to t_{n+1}} p(\mathcal{T}) \psi_\theta(t_n) h_n(\mathcal{T}) = 0, \quad \lim_{t_n \to t_{n-1}} p(\mathcal{T}) \psi_\theta(t_n) h_n(\mathcal{T}) = 0, \ \forall n \in [N_T],$$
$$\mathbb{E}[\psi_\theta^2(t_n) h_n(\mathcal{T})] < \infty, \quad \mathbb{E}[\psi_\theta(t_n) \frac{\partial h_n(\mathcal{T})}{\partial t_n}] < \infty, \forall n \in [N_T]. \tag{9}$$

One can verify that such weight functions are easy to find for most $p(\mathcal{T})$ and $\psi_\theta(t_n)$. With a valid weight function $\mathbf{h}$, the explicit WSM objective can be defined as:

$$\mathcal{L}_{\text{WSM}}(\theta) = \frac{1}{2} \mathbb{E}_{p(\mathcal{T})} \left[ \sum_{n=1}^{N_T} (\psi(t_n) - \psi_\theta(t_n))^2 h_n(\mathcal{T}) \right]. \tag{10}$$

The introduction of the weight function allows control over the values of the integrand at the boundaries of the integration domain, thereby eliminating the last two terms in Equation (8).

**Theorem 3.2.** *Assume the true intensity is in the family of the model intensity, denoted as $\lambda(t) = \lambda_{\theta^*}(t)$, where $\theta^* \in \Theta$. We further assume that $\frac{\partial \log \lambda_{\theta_1}(t)}{\partial t} = \frac{\partial \log \lambda_{\theta_2}(t)}{\partial t}$ a.s. gives $\theta_1 = \theta_2$. Then the unique minimizer of $\mathcal{L}_{WSM}(\theta)$ is $\theta^*$.*

The explicit WSM objective is not practical as it depends on the unknown data distribution $p(\mathcal{T})$, so we further derive the implicit WSM objective which is tractable.

**Theorem 3.3.** *Assume that all functions and expectations in $\mathcal{L}_{WSM}(\theta)$ and $\mathcal{J}_{WSM}(\theta)$ are well defined, Equation (9) is satisfied, we have,*

$$\mathcal{L}_{WSM}(\theta) = \mathcal{J}_{WSM}(\theta) + const,$$

$$\mathcal{J}_{WSM}(\theta) = \mathbb{E}_{p(\mathcal{T})}\left[\sum_{n=1}^{N_T} \frac{1}{2}\psi_\theta^2(t_n)h_n(\mathcal{T}) + \frac{\partial \psi_\theta(t_n)}{\partial t_n}h_n(\mathcal{T}) + \psi_\theta(t_n)\frac{\partial h_n(\mathcal{T})}{\partial t_n}\right]. \tag{11}$$

For general Poisson processes, Equation (11) is always valid with a suitable weight function. Thus, we do not need to worry about the issues of failure that may arise when using Equation (7).

## 4 Autoregressive Score Matching for Hawkes Processes

Similarly, we analyze the usage of ASM for Hawkes processes and its failure in achieving consistent estimation. Subsequently, we propose a provably consistent autoregressive WSM (AWSM) estimator.

### 4.1 Failure of Autoregressive Score Matching for Hawkes Process

The original SM, even when adjusted by a weight function, is not suitable for point processes with autoregressive structures, such as Hawkes process. Because in such cases, directly calculating the score still includes the intensity integral, which is precisely what the use of SM aims to avoid. Therefore, an ASM method is proposed for parameter estimation for Hawkes process in [10].

Consider a Hawkes process $\mathcal{T} = (t_1, \ldots, t_{N_T})$ on $[0, T]$ with the underlying conditional probability density of $t_n$ denoted as $p(t_n|\mathcal{F}_{t_{n-1}})$. The parameterized conditional probability density model of $t_n$ is $p_\theta(t_n|\mathcal{F}_{t_{n-1}})$. We denote the conditional score as $\psi(t_n|\mathcal{F}_{t_{n-1}}) = \frac{\partial}{\partial t_n}\log p(t_n|\mathcal{F}_{t_{n-1}}) = \frac{\partial}{\partial t_n}\log \lambda(t_n|\mathcal{F}_{t_{n-1}}) - \lambda(t_n|\mathcal{F}_{t_{n-1}}), n = 1, \ldots N_T$. An explicit ASM objective is defined as:

$$\mathcal{L}_{\text{ASM}}(\theta) = \frac{1}{2}\mathbb{E}_{p(\mathcal{T})}\left[\sum_{n=1}^{N_T}(\psi(t_n|\mathcal{F}_{t_{n-1}}) - \psi_\theta(t_n|\mathcal{F}_{t_{n-1}}))^2\right]. \tag{12}$$

Similarly, to make ASM practical, [10] assumed that specific regularity conditions are satisfied. Therefore, an implicit ASM is proposed accordingly:

$$\mathcal{J}_{\text{ASM}}(\theta) = \mathbb{E}_{p(\mathcal{T})}\left[\sum_{n=1}^{N_T}\frac{1}{2}\psi_\theta^2(t_n|\mathcal{F}_{t_{n-1}}) + \frac{\partial \psi_\theta(t_n|\mathcal{F}_{t_{n-1}})}{\partial t_n}\right]. \tag{13}$$

However, the same issue as in the Poisson process arises here. The regularity conditions required to eliminate the unknown data distribution do not hold. Therefore, we cannot derive the implicit ASM in Equation (13) based on the explicit ASM in Equation (12).

**Proposition 4.1.** *Assume that all functions and expectations in $\mathcal{L}_{ASM}(\theta)$ and $\mathcal{J}_{ASM}(\theta)$ are well defined, we have,*

$$\mathcal{L}_{ASM}(\theta) = \mathcal{J}_{ASM}(\theta) + const + \sum_{n=1}^{\infty}\int p(\mathcal{T}_{:n-1})p(t_n|\mathcal{F}_{t_{n-1}})\psi_\theta(t_n|\mathcal{F}_{t_{n-1}})\Big|_{t_n=T}d\mathcal{T}_{:n-1}$$

$$- \sum_{n=1}^{\infty}\int p(\mathcal{T}_{:n-1})p(t_n|\mathcal{F}_{t_{n-1}})\psi_\theta(t_n|\mathcal{F}_{t_{n-1}})\Big|_{t_n=t_{n-1}}d\mathcal{T}_{:n-1}. \tag{14}$$

*Therefore, $\mathcal{L}_{ASM}(\theta)$ is equivalent to $\mathcal{J}_{ASM}(\theta)$ if and only if the sum of last two terms is a constant not containing $\theta$.*

Generally speaking, for most Hawkes processes, the sum of the last two terms in Equation (14) still contains $\theta$, even for a common Hawkes process with an exponential decay triggering kernel. We illustrate this example in Section 6.2. This implies that $\mathcal{J}_{\text{ASM}}$ fails for general Hawkes processes.

## 4.2 Autoregressive Weighted Score Matching

Similarly, to address the situation where ASM fails, we introduce the AWSM for Hawkes process. We present the conditions that a valid weight function $\mathbf{h}$ should satisfy :

$$\lim_{t_n \to T} p(\mathcal{T}_{1:n})\psi_\theta(t_n|\mathcal{F}_{t_{n-1}})h_n(\mathcal{T}) = 0, \lim_{t_n \to t_{n-1}} p(\mathcal{T}_{1:n})\psi_\theta(t_n|\mathcal{F}_{t_{n-1}})h_n(\mathcal{T}) = 0, \ \forall n \in [N_T],$$

$$\mathbb{E}[\psi_\theta^2(t_n|\mathcal{F}_{t_{n-1}})h_n(\mathcal{T})] < \infty, \ \mathbb{E}[\psi_\theta(t_n|\mathcal{F}_{t_{n-1}})\frac{\partial h_n(\mathcal{T})}{\partial t_n}] < \infty, \forall n \in [N_T]. \tag{15}$$

With a valid weight function $\mathbf{h}$, the explicit AWSM objective can be defined as:

$$\mathcal{L}_{\text{AWSM}}(\theta) = \frac{1}{2}\mathbb{E}_{p(\mathcal{T})}\left[\sum_{n=1}^{N_T}(\psi(t_n|\mathcal{F}_{t_{n-1}}) - \psi_\theta(t_n|\mathcal{F}_{t_{n-1}}))^2 h_n(\mathcal{T})\right]. \tag{16}$$

**Theorem 4.2.** *Assume the true conditional density is in the family of the model conditional density, denoted as $p(t_n|\mathcal{F}_{t_{n-1}}) = p_{\theta^*}(t_n|\mathcal{F}_{t_{n-1}})$, where $\theta^* \in \Theta$. We further assume that $p_{\theta_1}(t_n|\mathcal{F}_{t_{n-1}}) = p_{\theta_2}(t_n|\mathcal{F}_{t_{n-1}})$ a.e. gives $\theta_1 = \theta_2$. Then the unique minimizer of $\mathcal{L}_{AWSM}(\theta)$ is $\theta^*$.*

The explicit AWSM objective is not practical as it depends on the unknown data distribution $p(t_n|\mathcal{F}_{t_{n-1}})$, so we further derive the implicit AWSM objective which is tractable.

**Theorem 4.3.** *Assume that all functions and expectations in $\mathcal{L}_{AWSM}(\theta)$ and $\mathcal{J}_{AWSM}(\theta)$ are well defined, Equation (15) are satisfied, we have,*

$$\mathcal{L}_{AWSM}(\theta) = \mathcal{J}_{AWSM}(\theta) + const,$$

$$\mathcal{J}_{AWSM}(\theta) = \mathbb{E}_{p(\mathcal{T})}\left[\sum_{n=1}^{N_T}\frac{1}{2}\psi_\theta^2(t_n|\mathcal{F}_{t_{n-1}})h_n(\mathcal{T}) + \frac{\partial\psi_\theta(t_n|\mathcal{F}_{t_{n-1}})}{\partial t_n}h_n(\mathcal{T}) + \psi_\theta(t_n|\mathcal{F}_{t_{n-1}})\frac{\partial h_n(\mathcal{T})}{\partial t_n}\right]. \tag{17}$$

For general Hawkes processes, Equation (17) is always valid with a suitable weight function. Thus, we do not need to worry about the issues of failure that may arise when using Equation (13).

**Multivariate Hawkes Processes**  For the multivariate case, events are $\{(t_1, k_1), \ldots, (t_{N_T}, k_{N_T})\}$ with $k_n \in 1, \ldots, K$ denoting the event type of the $n$-th event. The history up to the $(n-1)$-th event is denoted by $\mathcal{F}_{t_{n-1}}$. We need to consider both the distributions of event times and event types. For the temporal distribution, we use the AWSM objective with the temporal score $\psi(t_n|\mathcal{F}_{t_{n-1}}) = \frac{\partial}{\partial t_n}\log p(t_n|\mathcal{F}_{t_{n-1}})$ as before. For the type distribution, since we do not need to compute the intensity integral, we directly use the cross-entropy objective:

$$\mathcal{J}_{\text{CE}}(\theta) = \mathbb{E}_{p(\mathcal{T})}\left[\sum_{n=1}^{N_T}\log p_\theta(k_n|\mathcal{F}_{t_{n-1}}, t_n)\right] = \mathbb{E}\left[\sum_{n=1}^{N_T}\log\lambda_{k_n}(t_n|\mathcal{F}_{t_{n-1}}; \theta) - \log\lambda(t_n|\mathcal{F}_{t_{n-1}}; \theta)\right], \tag{18}$$

where $\lambda = \sum_{k=1}^{K}\lambda_k$. The final loss is $\mathcal{J}(\theta) = \mathcal{J}_{\text{AWSM}}(\theta) + \alpha\mathcal{J}_{\text{CE}}(\theta)$; $\alpha$ is a balancing coefficient.

## 5 Theoretical Analysis

In this section, we analyze the statistical properties of AWSM estimator of univariate Hawkes process. Similar conclusions also hold for the WSM estimator of Poisson process, as discussed in Appendix C.5. We consider $M$ i.i.d. sequences $\{t_1^{(m)}, \ldots, t_{N_m}^{(m)}\}_{m=1}^{M}$ from $p(\mathcal{T})$ of a Hawkes process. We assume the true density is in the family of the model density, denoted as $p(\mathcal{T}) = p_{\theta^*}(\mathcal{T})$, where $\theta^* \in \Theta \subset \mathbb{R}^r$. The estimate $\hat{\theta}$ is obtained by $\hat{\theta} = \arg\min_{\theta\in\Theta}\hat{\mathcal{J}}_{\text{AWSM}}(\theta)$ where $\hat{\mathcal{J}}_{\text{AWSM}}$ represents the empirical loss. Below we omit the subscript AWSM as it does not cause any ambiguity.

### 5.1 Asymptotic Property

We first establish the consistency of $\hat{\theta}$ for a Hawkes process.

**Theorem 5.1.** *Under mild regularity Assumptions C.1 to C.3, we have $\hat{\theta} \xrightarrow{p} \theta^*$ as $M \to \infty$.*

## 5.2 Non-asymptotic Error Bound

Then, we establish a non-asymptotic error bound for $\hat{\theta}$. We define

$$\mathcal{J}_{\mathbf{h}}(\theta) = \mathbb{E}_{p(\mathcal{T})} \left[ \sum_{n=1}^{N_T} \big[ \underbrace{\frac{1}{2}\psi_\theta^2(t_n|\mathcal{F}_{t_{n-1}}) + \frac{\partial\psi_\theta(t_n|\mathcal{F}_{t_{n-1}})}{\partial t_n}}_{A_n(\mathcal{T},\theta)} \big] h_n(\mathcal{T}) + \underbrace{\psi_\theta(t_n|\mathcal{F}_{t_{n-1}})}_{B_n(\mathcal{T},\theta)} \frac{\partial h_n(\mathcal{T})}{\partial t_n} \right].$$

**Assumption 5.2.** Assume there exists $\alpha > 1$ such that,

$$\inf_{\theta:||\theta-\theta^*||\geq\delta} \mathcal{J}_{\mathbf{h}}(\theta) - \mathcal{J}_{\mathbf{h}}(\theta^*) \geq C_{\mathbf{h}}\delta^\alpha$$

holds for any small $\delta$. Here, $C_{\mathbf{h}}$ is a positive constant that depends on the weight function $\mathbf{h}$ such that $C_{a\mathbf{h}} = aC_{\mathbf{h}}$ for any positive constant $a$. $\|\cdot\|$ is the euclidean norm.

**Assumption 5.3.** For $\forall n \in \mathbb{N}^+$, there exists $\dot{A}_n(\mathcal{T}), \dot{B}_n(\mathcal{T})$ such that,

$$|A_n(\mathcal{T},\theta_1) - A_n(\mathcal{T},\theta_2)| \leq \dot{A}_n(\mathcal{T})\|\theta_1 - \theta_2\|, \quad |B_n(\mathcal{T},\theta_1) - B_n(\mathcal{T},\theta_2)| \leq \dot{B}_n(\mathcal{T})\|\theta_1 - \theta_2\|.$$

**Theorem 5.4.** *Given that $\hat{\theta}$ converges to $\theta^*$ in probability, combined with Assumptions 5.2 and 5.3, for $\delta < CK_\alpha \frac{\sqrt{r}}{2^{\alpha-1}} \frac{\Gamma(\mathbf{h},A,B)}{C_{\mathbf{h}}}$, we have*

$$Pr\left[ \|\hat{\theta} - \theta^*\| \leq \left( CK_\alpha \frac{\Gamma(\mathbf{h},A,B)}{\delta C_{\mathbf{h}}} \sqrt{\frac{r}{M}} \right)^{-1/(\alpha-1)} \right] \geq 1 - \delta, \tag{19}$$

*where $\Gamma(\mathbf{h}, A, B) = \sqrt{\mathbb{E}_{p(\mathcal{T})}\left\{ \sum_{n=1}^{N_T} \big[(\dot{A}_n(\mathcal{T})h_n(\mathcal{T})) + (\dot{B}_n(\mathcal{T})\frac{\partial h_n(\mathcal{T})}{\partial t_n})\big] \right\}^2}$, $C$ is a universal constant, $K_\alpha = \frac{2^{2\alpha}}{2^{\alpha-1}-1}$, and $r$ is the number of dimensions of $\theta$.*

## 5.3 Discussion on Optimal Weight Function

In Sections 3 and 4, we only provide the conditions that the weight function needs to satisfy. In fact, there are many weight functions that satisfy these conditions. The optimal weight function should minimize the error bound in Equation (19), which is equivalent to minimizing the coefficient $\frac{\Gamma(\mathbf{h},A,B)}{C_{\mathbf{h}}}$. The numerator cannot be analytically computed as it involves an unknown distribution $p(\mathcal{T})$, but we can maximize the denominator $C_{\mathbf{h}}$ in a predefined function family.

**Theorem 5.5.** *Define $\mathbf{h}^0$ to be a weight function with its $n$-th element defined as the distance between $t_n$ and the boundary of its support $[t_{n-1}, T]$:*

$$h_n^0(t_n) = \frac{T - t_{n-1}}{2} - |t_n - (T + t_{n-1})/2|.$$

*We have,*

$$\mathbf{h}^0 \in \arg\max_{\mathbf{h}\in\mathcal{H}} \inf_{\theta:||\theta-\theta^*||\geq\delta} \mathcal{J}_{\mathbf{h}}(\theta) - \mathcal{J}_{\mathbf{h}}(\theta^*)$$

*where $\mathcal{H}$ is a family of functions that is rigorously defined in Equation (27).*

Combined with Assumption 5.2, it can be observed that $\mathbf{h}^0$ maximizes $C_{\mathbf{h}}$ in $\mathcal{H}$. Though it does not necessarily optimize $\frac{\Gamma(\mathbf{h},A,B)}{C_{\mathbf{h}}}$, it is an adequate choice without using any information on $p(\mathcal{T})$. We also discuss it heuristically in Appendix C.4. It is worth noting that $h_n^0$ is not continuously differentiable; however, it is weakly differentiable. Its weak derivative is continuous, allowing both integration by parts and statistical theory to hold. In subsequent experiments, we consistently employ this optimal weight function when $T$ is available or can be approximated for the dataset.

## 6 Experiments

In this section, we validate our proposed (A)WSM on parametric or deep point process models. For parametric models, we focus on verifying whether (A)WSM can accurately recover the ground-truth parameters. For deep point process models, we confirm that our new training method is also applicable to deep neural network models. [2]

---

[2]Experiments are performed using an NVIDIA A16 GPU, 15GB memory.

Table 1: The MAE of three models trained by MLE, (A)SM, and (A)WSM on the synthetic dataset. For the 2-variate processes, we only present the estimation results for some parameters here. The results for other parameters can be found in Table 3.

| ESTIMATOR | POISSON | EXP-HAWKES | | | GAUSSIAN-HAWKES | | |
|---|---|---|---|---|---|---|---|
| | $\phi$ | $\alpha_{11}$ | $\alpha_{12}$ | $\mu_1$ | $\alpha_{11}$ | $\mu_1$ | $\sigma$ |
| (A)WSM | $0.07_{\pm 0.14}$ | $0.041_{\pm 0.041}$ | $0.026_{\pm 0.001}$ | $\mathbf{0.011_{\pm 0.010}}$ | $0.153_{\pm 0.162}$ | $0.022_{\pm 0.023}$ | $0.060_{\pm 0.066}$ |
| (A)SM | $1.56_{\pm 0.01}$ | $1.600_{\pm 0.001}$ | $0.200_{\pm 14.30}$ | $0.700_{\pm 0.272}$ | $1.413_{\pm 0.263}$ | $0.696_{\pm 0.267}$ | $2.507_{\pm 1.957}$ |
| MLE | $\mathbf{-0.02_{\pm 0.10}}$ | $\mathbf{0.028_{\pm 0.015}}$ | $\mathbf{0.014_{\pm 0.002}}$ | $0.012_{\pm 0.006}$ | $\mathbf{0.098_{\pm 0.107}}$ | $\mathbf{0.017_{\pm 0.019}}$ | $\mathbf{0.051 \pm 0.049}$ |

## 6.1 Baselines and Metrics

We consider three baseline parameter estimators: (1) **MLE** (2) implicit **(A)SM** [18; 24; 10] (3) Denoising Score Matching (**DSM**) [10]. We briefly introduce DSM in deep point process models.

For deep Hawkes process training, DSM is employed as follows. For observed timestamps $t_n^{(m)}$ in $m$-th sequence, we sample $L$ noise samples $\tilde{t}_{n,l}^{(m)} = t_n^{(m)} + \epsilon_{n,l}^{(m)}, l = 1, \ldots, L$, where $\text{Var}(\varepsilon_{n,L}^{(m)}) = \sigma^2$ and get the DSM objective:

$$\hat{\mathcal{J}}(\theta) = \frac{1}{M} \sum_{m=1}^{M} \sum_{n=1}^{N_m} \sum_{l=1}^{L} \frac{1}{2L} [\psi_\theta(\tilde{t}_{n,l}^{(m)} | \mathcal{F}_{t_{n-1}^{(m)}}) + \frac{\varepsilon_{n,l}^{(m)}}{\sigma^2}] + \alpha \hat{\mathcal{J}}_{\text{CE}}(\theta),$$

where $\mathcal{J}_{\text{CE}}(\theta)$ is the cross-entropy loss defined in Equation (18).

To compare the performance of different methods, for parametric models on synthetic data, we use the mean absolute error (**MAE**, $|\hat{\theta} - \theta|$) between the ground-truth parameters and the estimates as a metric since the ground-truth parameters are known. For deep point process models, we use the test log-likelihood (**TLL**) and the event type prediction accuracy (**ACC**) on the test data as metrics.

## 6.2 Parametric Models

**Datasets**   We validate the effectiveness of (A)WSM using three sets of synthetic data. (1) **Poisson Process**: This dataset is simulated from an inhomogeneous Poisson process with an intensity function $\lambda(t) = \exp(\theta \sin(t))$ with $T = 2$, $\theta = 2$. (2) **Exponential Hawkes Processes**: This dataset is simulated from 2-variate Hawkes processes with exponential decay triggering kernels $g_{ij}(\tau) = \alpha_{ij} \exp(-5\tau)$, $\tau > 0$ with $T = 10$, $\mu_1 = \mu_2 = 1$, $\alpha_{11} = 1.6, \alpha_{12} = 0.2, \alpha_{21} = \alpha_{22} = 1$. (3) **Gaussian Hawkes Processes**: This dataset is simulated from 2-variate Hawkes processes with Gaussian decay triggering kernels $g_{ij}(\tau) = \frac{\alpha_{ij}}{\sqrt{2\pi}\sigma} \exp(-\frac{\tau^2}{2\sigma^2})$, $\tau > 0$ with $T = 10$, $\mu_1 = \mu_2 = 1$, $\alpha_{11} = 1.6, \alpha_{12} = 0.2, \alpha_{21} = \alpha_{22} = 1, \sigma = 1$.

**Training Protocol**   We assume that we know the ground-truth model but do not know its parameters. Therefore, we use the ground-truth model as the training model. The purpose is to verify whether the estimator can recover the ground-truth parameters. For each dataset, we collect a total of 1000 sequences. We run 500 iterations of gradient descent using Adam [8] as the optimizer for all scenarios. For MLE, the intensity integral is computed through numerical integration, with the number of integration nodes set to 100 to achieve a considerable level of accuracy. We change the random seed 3 times to compute the mean and standard deviation of MAE.

**Results**   In Table 1, we report the MAE of parameter estimates for three models trained by MLE, (A)SM, and (A)WSM on the synthetic dataset. We can see that both MLE and (A)WSM achieve small MAE on three types of data. However, the MAE of (A)SM is large. As we have theoretically demonstrated earlier, this is because MLE and (A)WSM estimators are consistent. In contrast, (A)SM, due to the absence of the required regularity conditions in the three cases, has an incomplete objective and cannot accurately estimate parameters. In Figure 1, we showcase the learned intensity functions. Both MLE and (A)WSM successfully captured the ground truth, while (A)SM fails.

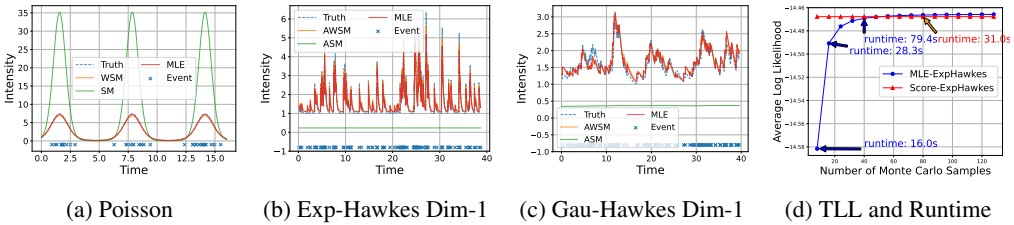

| (a) Poisson | (b) Exp-Hawkes Dim-1 | (c) Gau-Hawkes Dim-1 | (d) TLL and Runtime |

Figure 1: The learned intensity functions from MLE, (A)SM, and (A)WSM on (a) Poisson, (b) Exp-Hawkes and (c) Gaussian-Hawkes. We present the results for the 1-st dimension. The 2-nd dimension are in Appendix D. The ground truth, MLE, and (A)WSM nearly overlap, while (A)SM differs. (d) The TLL and runtime of (A)WSM and MLE w.r.t. the number of integration nodes.

## 6.3 Deep Point Processes Models

**Datasets**   We consider four real datasets. (1) **Half-Sin Hawkes Process**: This is a synthetic 2-variate Hawkes process with trigerring kernel $g_{ij} = \alpha_{ij} sin(\tau), \tau \in (0, \pi)$, $K = 2$. (2) **StackOverflow** [7]: This dataset has two years of user awards on StackOverflow. Each user received a sequence of badges and there are $K = 22$ kinds of badges. (3) **Retweet** [25]: This dataset includes sequences indicating how each novel tweets are forwarded by other users. Retweeter categories serve as event types $K = 3$. (4) **Taobao** [21]: This dataset comprises user activities on Taobao (in total $K = 17$ event types). For each dataset, we follow the default training/dev/testing split in the repository.

**Training Protocol**   In recent years, many deep point process models have been proposed. Here, we focus on two of the most popular attention-based Hawkes process models: **SAHP** [23] and **THP** [27]. We deploy AWSM and ASM on THP and SAHP. For each dataset, we train 3 seeds with the same epochs and report the mean and standard deviation of the best TLL and ACC. When using MLE, we adopt numerical integration to calculate the intensity integral. To ensure model accuracy, the number of integration nodes is set to be large enough as we sample 10 nodes between every two adjacent events. When using DSM, we tune the variance of noise for better results. When using AWSM, since for real datasets, the true observation endpoint $T$ is unknown. We choose the maximum event time of each batch as the observation endpoint for weight function $\boldsymbol{h}^0$. This may lead to unsatisfying results since real datasets may not be sampled during a unified time window. We provide a remedy for this as discussed in Appendix D.1. Details of training and testing hyperparameters are provided in Appendix D.2.

**Results**   In Table 2, we report the performance of SAHP and THP trained using three different methods, namely MLE, AWSM, and DSM, on four datasets. It is evident from the results that models trained with MLE and AWSM exhibit very similar performance in terms of both TLL and ACC on the test data. This indicates consistency between MLE and AWSM, as they yield comparable model parameters. For DSM, it is significantly inferior to the performance of MLE and AWSM. This may result from the fact that the DSM objective is a biased estimation of the original SM objective and fails to produce consistent estimation when $\sigma > 0$ as discussed in [20]. For ASM, it completely fails in the scenarios mentioned above. It is unable to estimate the correct parameters, and its results are not reported. Generally, for complex point process models such as deep Hawkes processes, the necessary regularity conditions are not satisfied, meaning that ASM's objective is incomplete.

## 6.4 Advantage of (A)WSM over MLE

The key advantage of (A)WSM over MLE is its avoidance of computing intensity integrals, which can be computationally intensive for complex point process models and impact MLE accuracy. We evaluate the test log-likelihood of MLE and AWSM on the Exp-Hawkes dataset as the number of integration nodes varies. As shown in Figure 1d, with a limited number of nodes, MLE is faster but exhibits substantial estimation errors. Increasing the number of nodes reduces the error but significantly increases computation time. In this scenario, AWSM is much faster than MLE with the same accuracy, thus offering better computational efficiency.

Table 2: The TLL and ACC of two attention-based deep Hawkes process models trained by MLE and AWSM on four datasets. Because ASM estimator completely fails, we do not report its results.

| Dataset | SAHP (TLL↑) | | | THP (TLL↑) | | |
|---|---|---|---|---|---|---|
| | MLE | AWSM | DSM | MLE | AWSM | DSM |
| Half-Sin | $1.542_{\pm0.038}$ | $\mathbf{1.703_{\pm0.014}}$ | $0.804_{\pm0.353}$ | $1.161_{\pm0.031}$ | $\mathbf{1.271_{\pm0.036}}$ | $-0.385_{\pm0.033}$ |
| Stackoverflow | $\mathbf{-2.428_{\pm0.14}}$ | $-2.541_{\pm0.461}$ | $-2.629_{\pm0.068}$ | $\mathbf{-2.368_{\pm0.003}}$ | $-2.508_{\pm0.007}$ | $-2.782_{\pm0.034}$ |
| Tabao | $\mathbf{-1.050_{\pm0.100}}$ | $-1.373_{\pm0.091}$ | $-1.911_{\pm0.049}$ | $-1.052_{\pm0.012}$ | $\mathbf{-0.948_{\pm0.004}}$ | $-1.791_{\pm0.040}$ |
| Retweets | $\mathbf{0.454_{\pm0.009}}$ | $0.411_{\pm0.077}$ | $0.110_{\pm0.186}$ | $\mathbf{0.421_{\pm0.012}}$ | $0.419_{\pm0.009}$ | $-0.183_{\pm0.197}$ |

| Dataset | SAHP (ACC↑) | | | THP (ACC↑) | | |
|---|---|---|---|---|---|---|
| | MLE | AWSM | DSM | MLE | AWSM | DSM |
| Half-Sin | $0.502_{\pm0.001}$ | $\mathbf{0.505_{\pm0.001}}$ | $0.501_{\pm0.001}$ | $0.508_{\pm0.016}$ | $\mathbf{0.523_{\pm0.010}}$ | $0.503_{\pm0.001}$ |
| Stackoverflow | $0.461_{\pm0.001}$ | $\mathbf{0.462_{\pm0.01}}$ | $0.421_{\pm0.042}$ | $0.461_{\pm0.001}$ | $\mathbf{0.462_{\pm0.001}}$ | $0.445_{\pm0.016}$ |
| Tabao | $\mathbf{0.572_{\pm0.022}}$ | $0.455_{\pm0.011}$ | $0.421_{\pm0.017}$ | $\mathbf{0.594_{\pm0.001}}$ | $0.592_{\pm0.002}$ | $0.435_{\pm0.010}$ |
| Retweets | $0.454_{\pm0.009}$ | $0.411_{\pm0.077}$ | $0.590_{\pm0.009}$ | $\mathbf{0.594_{\pm0.001}}$ | $0.592_{\pm0.002}$ | $0.556_{\pm0.011}$ |

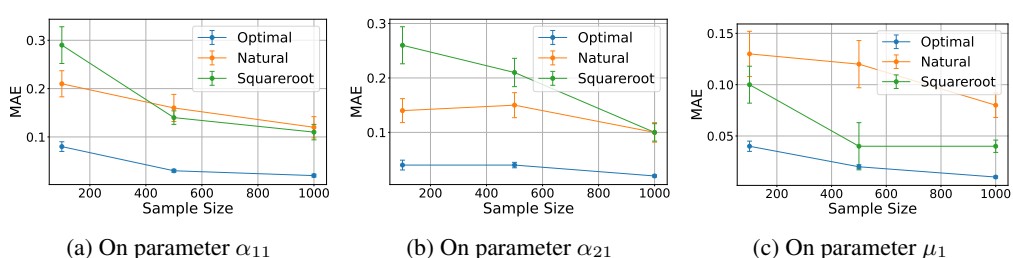

(a) On parameter $\alpha_{11}$     (b) On parameter $\alpha_{21}$     (c) On parameter $\mu_1$

Figure 2: MAE of parameter estimation versus sample size for three different weight functions on Exponential-Hawkes Model. Our near-optimal weight function outperforms the rest two valid weight functions in all sample sizes. We only show results for three parameters. The rest parameters have almost the same paradigm.

## 6.5 Comparison Between Weights

Though we provide theoretical insight into the choice of an optimal weight function for AWSM, its validity still needs to be testified by experiments. Here, we compare the near-optimal weight $\boldsymbol{h}^0$ with natural weight $\boldsymbol{h}^1$ and squareroot weight $\boldsymbol{h}^2$ satisfying Equation (15),

$$h_n^1(t_n) = (t_n - t_{n-1})(T - t_n), h_n^2(t_n) = \sqrt{(t_n - t_{n-1})(T - t_n)}.$$

All three weight functions can be applied in AWSM to recover ground-truth parameters, however with different convergence rates. We carry out experiments on synthetic data for exponential-decay model with the same setting as Section 6.2 in our paper. We measure their MAE for different sample sizes in Figure 2 and find that $\boldsymbol{h^0}$ does achieve the best results among the three weight functions.

## 7 Limitations

The current limitation of the methodology is that some real data are collected from multiple time intervals $[0, T_1], \ldots, [0, T_L]$ or collated in a fixed time interval $[0, T]$ with unknown $T$. However, for a score matching to be valid, the required weight function must involve knowledge of $T$. Currently, our remedy including approximate $T$ or performing data truncation as discussed in Appendix D.1.

## 8 Conclusions

In conclusion, the SM estimator for point processes can overcome the challenges associated with intensity integrals in MLE. While existing works have proposed SM estimators for point processes, our investigation reveals that they prove effective only for specific problems and fall short in more general cases. To address this issue, our work introduces a novel approach: the (A)WSM estimator for point processes, offering both theoretical soundness and empirical success.

## Acknowledgments and Disclosure of Funding

This work was supported by NSFC Project (No. 62106121), the MOE Project of Key Research Institute of Humanities and Social Sciences (22JJD110001), the fundamental research funds for the central universities, and the research funds of Renmin University of China (24XNKJ13).

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

# A  Proof of Results in Section 3

## A.1  Proof of Proposition 3.1

*Proof.* First consider the cross term in $\mathcal{L}_{\mathrm{SM}}(\theta)$ and expand the expectation,

$$
\mathbb{E}_{p(\mathcal{T})}\left[\sum_{n=1}^{N_T}\frac{\partial \log p(\mathcal{T})}{\partial t_n}\frac{\partial \log p_\theta(\mathcal{T})}{\partial t_n}\right]
$$

$$
= \sum_{N=1}^{\infty}\int p(t_1,\ldots,t_N)\sum_{n=1}^{N}\frac{\partial \log p(t_1,\ldots,t_N)}{\partial t_n}\frac{\partial \log p_\theta(t_1,\ldots,t_N)}{\partial t_n}d\mathcal{T}_{1:N}
$$

$$
= \sum_{N=1}^{\infty}\int \sum_{n=1}^{N}\frac{\partial p(t_1,\ldots,t_N)}{\partial t_n}\frac{\partial \log p_\theta(t_1,\ldots,t_N)}{\partial t_n}d\mathcal{T}_{1:N}
\tag{20}
$$

$$
= \sum_{N=1}^{\infty}\left\{\int \sum_{n=1}^{N}[p(t_1,\ldots,t_N)\frac{\partial \log p_\theta(t_1,\ldots,t_N)}{\partial t_n}]|_{t_n=t_{n-1}}^{t_n=t_{n+1}}d\mathcal{T}_{-n}\right.
$$

$$
\left. - \int \sum_{n=1}^{N}p(t_1,\ldots,t_N)\frac{\partial^2 \log p_\theta(t_1,\ldots,t_N)}{\partial t_n^2}d\mathcal{T}_{1:N}\right\}.
$$

The third equation uses an integral-by-part trick. All integrations above are taken within the area of $\{0 \le t_1 \le \ldots \le t_N \le T\}$ or $\{0 \le t_1 \le \ldots \le t_{n-1} \le t_{n+1} \le \ldots \le t_N \le T\}$ when $t_n$ has been integrated. For the first term in the right side of the last equation, notice that,

$$
\frac{\partial \log p_\theta(t_1,\ldots,t_N)}{\partial t_N} = \frac{\partial}{\partial t_n}\log \lambda_\theta(t_n),\quad \frac{\partial}{\partial t_n}\log \lambda_\theta(t_n)\big|_{t_n=t_{n-1}} = \frac{\partial}{\partial t_{n-1}}\log \lambda_\theta(t_{n-1})|_{t_{n-1}=t_n}.
\tag{21}
$$

Therefore, we can see that, for $n \in \{2,\ldots,n\}$,

$$
\int [p(t_1,\ldots,t_N)\frac{\partial \log p_\theta(t_1,\ldots,t_N)}{\partial t_{n-1}}]\big|_{t_{n-1}=t_n}d\mathcal{T}_{-(n-1)}
$$

$$
= \int [p(t_1,\ldots,t_N)\frac{\partial \log p_\theta(t_1,\ldots,t_N)}{\partial t_n}]\big|_{t_n=t_{n-1}}d\mathcal{T}_{-n}.
$$

Using the above equation, we manage to cancel out most of the terms being summed in the right side of the thrid equation in Equation (20) and only leave the first and last term, which completes the proof.

$\square$

## A.2  Proof of Theorem 3.2

*Proof.* First, since $\mathcal{L}_{\mathrm{WSM}}(\theta) \ge 0$ and $\mathcal{L}_{\mathrm{WSM}}(\theta^*) = 0$, we see $\theta^*$ is a minimizer. If there exists another minimizer $\theta_1$, then we have

$$
\mathcal{L}_{\mathrm{WSM}}(\theta) = \sum_{N=1}^{\infty}\int \sum_{n=1}^{N}\left[\frac{\partial}{\partial t_n}\log \lambda_{\theta^*}(t_n) - \frac{\partial}{\partial t_n}\log \lambda_{\theta_1}(t_n)\right]^2 h_n(\mathcal{T})d\mathcal{T}.
$$

By the definition of $\mathbf{h}(\mathcal{T})$, for any $N$, since $\mathbf{h}(\mathcal{T}) > 0$ *a.s.* elementwisely on $\{0 \le t_1 \le \ldots \le t_N \le T\}$. So $\mathcal{L}_{\mathrm{WSM}}(\theta) = 0$ implies $\frac{\partial \log \lambda_{\theta^*}(t)}{\partial t} = \frac{\partial \log \lambda_{\theta_1}(t)}{\partial t}$ *a.e.* on $[0, T]$. By assumption, this implies $\theta_1 = \theta^*$, which completes the proof. $\square$

## A.3 Proof of Theorem 3.3

*Proof.* The proof is basically the same as the proof of Proposition 3.1. We first expand the expectation and consider the cross-term in $\mathcal{L}_{\text{WSM}}(\theta)$,

$$\mathbb{E}_{p(\mathcal{T})}\left[\sum_{n=1}^{N_T} \frac{\partial \log p(\mathcal{T})}{\partial t_n} \frac{\partial \log p_\theta(\mathcal{T})}{\partial t_n} h_n(\mathcal{T})\right] = \sum_{N=1}^{\infty} \int \sum_{n=1}^{N} \frac{\partial p(\mathcal{T})}{\partial t_n} \frac{\partial \log p_\theta(\mathcal{T})}{\partial t_n} h_n(\mathcal{T}) d\mathcal{T}_{1:N}$$

$$= \sum_{N=1}^{\infty} \left\{ \int \sum_{n=1}^{N} p(t_1, \ldots, t_N) \frac{\partial \log p_\theta(\mathcal{T})}{\partial t_n} h_n(\mathcal{T})\Big|_{t_n=t_{n-1}}^{t_n=t_{n+1}} d\mathcal{T}_{-n} \right. \tag{22}$$

$$\left. - \int p(t_1, \ldots, t_N) \sum_{n=1}^{N} \left[ \frac{\partial^2 \log p_\theta(\mathcal{T})}{\partial t_n^2} h_n(\mathcal{T}) + \frac{\partial \log p_\theta(\mathcal{T})}{\partial t_n} \frac{\partial h_n(\mathcal{T})}{\partial t_n} \right] d\mathcal{T}_{1:N} \right\}.$$

We denote $t_{N+1} = T$ and $t_0 = 0$ here. Using the first two equations in Equation (9), we have:

$$\int [p(t_1, \ldots, t_N) \frac{\partial \log p_\theta(t_1, \ldots, t_N)}{\partial t_n} h_n(\mathcal{T})]\big|_{t_n=t_{n+1}} d\mathcal{T}_{-n} = 0, \forall n \in [N],$$

$$\int [p(t_1, \ldots, t_N) \frac{\partial \log p_\theta(t_1, \ldots, t_N)}{\partial t_n} h_n(\mathcal{T})]\big|_{t_n=t_{n-1}} d\mathcal{T}_{-n} = 0, \forall n \in [N].$$

Therefore, the first intractable summation term in Equation (22) will disappear, and the second term equals $-\mathbb{E}_{p(\mathcal{T})}\left[\sum_{n=1}^{N_T} \frac{\partial}{\partial t_n}\psi_\theta(t_n)h_n(\mathcal{T}) + \psi_\theta(t_n)\frac{\partial}{\partial t_n}h_n(\mathcal{T})\right]$. The existence of such an expectation is due to the last two equations in Equation (9). Therefore, we complete the proof.

$\square$

We can see from the proof that, in Equation (9), the first two equations ensure that the integration by parts trick does not produce an intractable term, and the last two equations are simply regularity conditions that ensure all terms are well-defined.

## B  Proof of Results in Section 4

**Lemma B.1.** *Let $f(t_n, \mathcal{F}_{t_{n-1}})$ be a function of $t_n, \mathcal{F}_{t_{n-1}}$, where $n \in \{1, \ldots, N_T\}$. Then we have*

$$\mathbb{E}_{p(\mathcal{T})}\left[\sum_{n=1}^{N_T} f(t_n, \mathcal{F}_{t_{n-1}})\right] = \sum_{n=1}^{\infty} \int p(t_1, \ldots, t_n) f(t_n, \mathcal{F}_{t_{n-1}}) d\mathcal{T}_{1:n},$$

*where $p(t_1, \ldots, t_n)$ is the density of observing these timestamps, the integration is taken over $\{0 \le t_1 \le \ldots \le t_n \le T\}$.*

*Proof.* We first expand the expectation and obtain,

$$\mathbb{E}_{p(\mathcal{T})}\left[\sum_{n=1}^{N_T} f(t_n, \mathcal{F}_{t_{n-1}})\right] = \sum_{N=1}^{\infty} \int p(t_1, \ldots, t_N) \sum_{n=1}^{N} f(t_n, \mathcal{F}_{t_{n-1}}) d\mathcal{T}$$

$$= \sum_{N=1}^{\infty} \left\{ \int \int \sum_{n=1}^{N-1} p(t_1, \ldots, t_n) p(t_{n+1}, \ldots, t_N | \mathcal{F}_{t_n}) f(t_n, \mathcal{F}_{t_{n-1}}) d\mathcal{T}_{n+1:N} d\mathcal{T}_{1:n} \right.$$

$$\left. + \int p(t_1, \ldots, t_N) p(N_T = N | \mathcal{F}_{t_N}) f(t_N, \mathcal{F}_{t_{N-1}}) d\mathcal{T} \right\}.$$

At this point, we need to first integrate out $t_{n+1}, \ldots, t_N$, which uses

$$\int p(t_{n+1}, \ldots, t_N | \mathcal{F}_{t_n}) d\mathcal{T}_{n+1:N} = p(N_T = N | \mathcal{F}_{t_n}).$$

Plug this in and we obtain,

$$\mathbb{E}_{p(\mathcal{T})}\left[\sum_{n=1}^{N_T} f(t_n, \mathcal{F}_{t_{n-1}})\right] = \sum_{N=1}^{\infty}\int\sum_{n=1}^{N} p(t_1,\dots,t_n)p(N_T=N|\mathcal{F}_{t_n})f(t_n,\mathcal{F}_{t_{n-1}})d\mathcal{T}_{1:n}$$

$$= \sum_{N=1}^{\infty}\sum_{n=1}^{N}\int p(t_1,\dots,t_n)p(N_T=N|\mathcal{F}_{t_n})f(t_n,\mathcal{F}_{t_{n-1}})d\mathcal{T}_{1:n}$$

$$= \sum_{n=1}^{\infty}\int p(t_1,\dots,t_n)f(t_n,\mathcal{F}_{t_{n-1}})\sum_{N=n}^{\infty}p(N_T=N|\mathcal{F}_{t_n})d\mathcal{T}_{1:n}$$

$$= \sum_{n=1}^{\infty}\int p(t_1,\dots,t_n)f(t_n,\mathcal{F}_{t_{n-1}})d\mathcal{T}_{1:n}.$$

The thrid equation adopts the exchange of summation. The feasibility is ensured by the assumption that the expectation in the left side of the equation exists and Fubini's theorem. The fourth equation use the fact that $\sum_{N=n}^{\infty}p(N_T=N|\mathcal{F}_{t_n})=1$.

$\square$

## B.1 Proof of Proposition 4.1

*Proof.* We use Lemma B.1 to the cross term of $\mathcal{L}_{\mathrm{ASM}}(\theta)$ and obtain,

$$\mathbb{E}_{p(\mathcal{T})}\left[\sum_{n=1}^{N_T}\psi(t_n|\mathcal{F}_{t_{n-1}})\psi_\theta(t_n|\mathcal{F}_{t_{n-1}})\right]$$

$$= \sum_{n=1}^{\infty}\int p(t_1,\dots t_n)\psi(t_n|\mathcal{F}_{t_{n-1}})\psi_\theta(t_n|\mathcal{F}_{t_{n-1}})d\mathcal{T}_{1:n}$$

$$= \int_{t_0}^{T} p(t_1|\mathcal{F}_{t_0})\frac{\partial\log p(t_1|\mathcal{F}_{t_0})}{\partial t_1}\psi_\theta(t_1|\mathcal{F}_{t_0})dt_1$$

$$+ \sum_{n=2}^{\infty}\int p(t_1,\dots,t_{n-1})p(t_n|\mathcal{F}_{t_{n-1}})\frac{\partial\log p(t_n|\mathcal{F}_{t_{n-1}})}{\partial t_n}\psi_\theta(t_n|\mathcal{F}_{t_{n-1}})d\mathcal{T}_{1:n}$$

$$= \int_{t_0}^{T}\frac{\partial p(t_1|\mathcal{F}_{t_0})}{\partial t_1}\psi_\theta(t_1|\mathcal{F}_{t_0})dt_1 + \sum_{n=2}^{\infty}\int\int_{t_{n-1}}^{T} p(t_1,\dots,t_{n-1})\frac{\partial p(t_n|\mathcal{F}_{t_{n-1}})}{\partial t_n}\psi_\theta(t_n|\mathcal{F}_{t_{n-1}})dt_n d\mathcal{T}_{1:n-1}$$

$$= p(t_1|\mathcal{F}_{t_0})\psi_\theta(t_1|\mathcal{F}_{t_0})\Big|_{t_1=t_0}^{t_1=T} - \int_{t_0}^{T} p(t_1|\mathcal{F}_{t_0})\frac{\partial\psi_\theta(t_1|\mathcal{F}_{t_0})}{\partial t_1}dt_1$$

$$+ \sum_{n=2}^{\infty}\int p(t_1,\dots,t_{n-1})p(t_n|\mathcal{F}_{t_{n-1}})\psi_\theta(t_n|\mathcal{F}_{t_{n-1}})\Big|_{t_n=t_{n-1}}^{t_n=T}d\mathcal{T}_{1:n-1}$$

$$- \sum_{n=2}^{\infty}\int p(t_1,\dots,t_n)\frac{\partial\psi_\theta(t_n|\mathcal{F}_{t_{n-1}})}{\partial t_n}d\mathcal{T}_{1:n}$$

$$= \sum_{n=1}^{\infty}\int p(\mathcal{T}_{:n-1})p(t_n|\mathcal{F}_{t_{n-1}})\psi_\theta(t_n|\mathcal{F}_{t_{n-1}})\Big|_{t_n=t_{n-1}}^{t_n=T}d\mathcal{T}_{:n-1} - \sum_{n=1}^{\infty}\int p(t_1,\dots,t_n)\frac{\partial\psi_\theta(t_n|\mathcal{F}_{t_{n-1}})}{\partial t_n}d\mathcal{T}_{1:n}$$

$$= \sum_{n=1}^{\infty}\int p(\mathcal{T}_{:n-1})p(t_n|\mathcal{F}_{t_{n-1}})\psi_\theta(t_n|\mathcal{F}_{t_{n-1}})\Big|_{t_n=t_{n-1}}^{t_n=T}d\mathcal{T}_{:n-1} - \mathbb{E}_{p(\mathcal{T})}\left[\sum_{n=1}^{N_T}\frac{\partial\psi_\theta(t_n|\mathcal{F}_{t_{n-1}})}{\partial t_n}\right].$$

For the fifth equation, we simply rearrange the terms. We recall that the notation $p(\mathcal{T}_{:0})$ equals one. For the last equation, we use Lemma B.1 again. This will be sufficient to complete the proof. $\square$

## B.2 Proof of Theorem 4.2

*Proof.* First, since $\mathcal{L}_{\text{AWSM}}(\theta) \geq 0$ and $\mathcal{L}_{\text{AWSM}}(\theta^*) = 0$, we see $\theta^*$ is a minimizer. If there exists another minimizer $\theta_1$, then we have

$$\mathcal{L}_{\text{AWSM}}(\theta_1) = \frac{1}{2} \sum_{N=1}^{\infty} \int \sum_{n=1}^{N} [\psi_{\theta^*}(t_n|\mathcal{F}_{t_{n-1}}) - \psi_{\theta_1}(t_n|\mathcal{F}_{t_{n-1}})]^2 h_n(\mathcal{T}) d\mathcal{T} = 0.$$

By the definition of $h_n(\mathcal{T})$, we have $h_n(\mathcal{T}) > 0$, $a.s.$ on $0 \leq t_1 \leq \ldots, \leq t_N \leq T$ for any $n \leq N$ and $N \in \mathbb{N}_+$. This implies, for any $n \leq N$ and $N \in \mathbb{N}_+$, $\psi_{\theta^*}(t_n|\mathcal{F}_{t_{n-1}}) = \psi_{\theta_1}(t_n|\mathcal{F}_{t_{n-1}})$, $a.e.$ on $\{0 \leq t_{n-1} \leq t_n \leq T\}$. Therefore we have

$$\log p_{\theta^*}(t_n|\mathcal{F}_{t_{n-1}}) = \log p_{\theta_1}(t_n|\mathcal{F}_{t_{n-1}}) + C, a.e.,$$

And we conclude that $C = 0$ since $\int_{t_{n-1}}^{T} p_\theta(t_n|\mathcal{F}_{t_{n-1}}) dt_n = 1$. Then by assumption, we have $p_{\theta^*}(t_n|\mathcal{F}_{t_{n-1}}) = p_{\theta_1}(t_n|\mathcal{F}_{t_{n-1}})$ $a.e. \Rightarrow \theta_1 = \theta^*$. $\qquad\square$

## B.3 Proof of Theorem 4.3

The proof is basically the same as the proof of Proposition 4.1. We first use the Lemma B.1 to the cross term of $\mathcal{L}_{\text{AWSM}}(\theta)$,

$$\mathbb{E}_{p(\mathcal{T})} \left[ \sum_{n=1}^{N_T} \psi(t_n|\mathcal{F}_{t_{n-1}}) \psi_\theta(t_n|\mathcal{F}_{t_{n-1}}) h_n(\mathcal{T}) \right]$$

$$= \sum_{n=1}^{\infty} \int p(t_1, \ldots t_n) \psi(t_n|\mathcal{F}_{t_{n-1}}) \psi_\theta(t_n|\mathcal{F}_{t_{n-1}}) h_n(\mathcal{T}) d\mathcal{T}_{1:n}$$

$$= \sum_{n=1}^{\infty} \int p(\mathcal{T}_{:n-1}) p(t_n|\mathcal{F}_{t_{n-1}}) \psi_\theta(t_n|\mathcal{F}_{t_{n-1}}) h_n(\mathcal{T}) \Big|_{t_n=t_{n-1}}^{t_n=T} d\mathcal{T}_{:n-1}$$

$$- \sum_{n=1}^{\infty} \int p(t_1, \ldots, t_n) \left[ \frac{\partial \psi_\theta(t_n|\mathcal{F}_{t_{n-1}})}{\partial t_n} h_n(\mathcal{T}) + \psi_\theta(t_n|\mathcal{F}_{t_{n-1}}) \frac{\partial h_n(\mathcal{T})}{\partial t_n} \right] d\mathcal{T}_{1:n}.$$

Between the second and the third line above, we omit the steps used in the derivation of Proposition 4.1 to make it concise. For the term in the third line above, it will be eliminated using Equation (15). For the term in the fourth line above, using Lemma B.1, we have:

$$- \sum_{n=1}^{\infty} \int p(t_1, \ldots, t_n) \left[ \frac{\partial \psi_\theta(t_n|\mathcal{F}_{t_{n-1}})}{\partial t_n} h_n(\mathcal{T}) + \psi_\theta(t_n|\mathcal{F}_{t_{n-1}}) \frac{\partial h_n(\mathcal{T})}{\partial t_n} \right] d\mathcal{T}_{1:n} =$$

$$- \mathbb{E}_{p(\mathcal{T})} \left[ \sum_{n=1}^{N_T} \frac{\partial \psi_\theta(t_n|\mathcal{F}_{t_{n-1}})}{\partial t_n} h_n(\mathcal{T}) + \psi_\theta(t_n|\mathcal{F}_{t_{n-1}}) \frac{\partial h_n(\mathcal{T})}{\partial t_n} \right].$$

The existence of the expectation is ensured by the last two terms in Equation (15).

# C Proof of Results in Section 5

We present all the regularity condition needed for establishing the consistency of our estimator:

**Assumption C.1.** The parameter space $\Theta$ is a compact set in $\mathbb{R}^d$ and contains an open set which contains $\theta^*$.

**Assumption C.2.** Both the $\mu_\theta(t)$ and $g_\theta(t)$ are twice continuously differentiable w.r.t. $t$ for all $\theta \in \Theta$ and those derivatives are continuous w.r.t. $\theta$.

We remind readers that $\mu_\theta(t)$ and $g_\theta(t)$ are the mean intensity function and the triggering kernel for a Hawkes process, first defined in Equation (2).

**Assumption C.3.** If both $\mu_{\theta_1}(t) = \mu_{\theta_2}(t)$ $a.s.$ and $g_{\theta_1}(t) = g_{\theta_2}(t)$ $a.s.$ on $[0, T]$, then $\theta_1 = \theta_2$.

## C.1 Proof of Theorem 5.1

*Proof.* As shown in Theorem 5.7 in [19], if we can prove a uniform in probability convergence for $\hat{\mathcal{J}}_{\text{AWSM}}(\theta)$ to $\mathcal{J}_{\text{AWSM}}(\theta)$, then using the fact that $\theta^*$ is a unique minimizer of $\mathcal{J}_{\text{WSM}}(\theta)$ in a compact set in $\mathbb{R}^d$, the consistency is proved. Therefore, we only prove the uniform in probability convergence.

For any $\theta \in \Theta$, for each sampled sequence $(t_1^{(m)}, \ldots, t_{N_m}^{(m)})$, we perceive the following value as a random variable,

$$
\xi_m = \sum_{n=1}^{N_m} \left[ \frac{1}{2} \psi_\theta(t_n^{(m)} | \mathcal{F}_{t_{n-1}^{(m)}}) h_n(\mathcal{T}^{(m)}) + \frac{\partial}{\partial t_n} \psi_\theta(t_n^{(m)} | \mathcal{F}_{t_{n-1}^{(m)}}) h_n(\mathcal{T}^{(m)}) \right.
$$
$$
\left. + \psi_\theta(t_n^{(m)} | \mathcal{F}_{t_{n-1}^{(m)}}) \frac{\partial}{\partial t_n} h_n(\mathcal{T}^{(m)}) \right].
$$

One can verify that it is a measurable map from the sample space to the real line, therefore indeed a random variable. And its expectation is $\mathcal{J}_{\text{AWSM}}(\theta)$, which is finite by assumption. Since different sequences are i.i.d. samples with finite expectation, using the weak law of large numbers, we have:

$$
\hat{\mathcal{J}}_{\text{AWSM}}(\theta) = \frac{1}{M} \sum_{m=1}^{M} \xi_m \xrightarrow{p} \mathcal{J}_{\text{AWSM}}(\theta), \forall \theta \in \Theta. \tag{23}
$$

Now we prove that this convergence is uniform in $\Theta$. Similar to [2], we first prove that

$$
\forall \varepsilon > 0, \exists \delta > 0, s.t. \forall \|\theta_1 - \theta_2\| < \delta, \mathbb{P}(|\hat{\mathcal{J}}_{\text{AWSM}}(\theta_1) - \hat{\mathcal{J}}_{\text{AWSM}}(\theta_2)| > \frac{1}{3}\varepsilon) \to 0, M \to \infty. \tag{24}
$$

First, by Assumption C.2, we know $\lambda_\theta(t_n | \mathcal{F}_{t_{n-1}})$ and $\frac{\partial}{\partial t_n} \lambda_\theta(t_n | \mathcal{F}_{t_{n-1}})$ are continuous w.r.t $\theta$. Therefore $\psi_\theta(t_n | \mathcal{F}_{t_{n-1}}) = \frac{\partial}{\partial t_n} \log \lambda_\theta(t_n | \mathcal{F}_{n-1}) - \lambda_\theta(t_n | \mathcal{F}_{t_{n-1}})$ and $\frac{\partial}{\partial t_n} \psi_\theta(t_n | \mathcal{F}_{t_{n-1}}) = \frac{\partial^2}{\partial t_n^2} \log \lambda_\theta(t_n | \mathcal{F}_{t_{n-1}}) - \frac{\partial}{\partial t_n} \lambda_\theta(t_n | \mathcal{F}_{t_{n-1}})$ are both continuous w.r.t. $\theta$, therefore the $\mathcal{J}_{\text{AWSM}}(\theta)$ is a continuous function of $\theta$. Since $\Theta$ is a compact set in $\mathbb{R}^d$, we know $\mathcal{J}_{\text{AWSM}}(\theta)$ is uniform continuous.

Therefore, we can bound $|\mathcal{J}_{\text{AWSM}}(\theta_1) - \mathcal{J}_{\text{AWSM}}(\theta_2)|$ using $\|\theta_1 - \theta_2\|$. Using the result in Equation (23), we know that for any $\varepsilon > 0$, we can find a uniform $\delta$ so that $|\mathcal{J}_{\text{AWSM}}(\theta_1) - \mathcal{J}_{\text{AWSM}}(\theta_2)| < \frac{1}{6}\varepsilon, \forall \|\theta_1 - \theta_2\| < \delta$. So that,

$$
\mathbb{P}(|\hat{\mathcal{J}}_{\text{AWSM}}(\theta_1) - \hat{\mathcal{J}}_{\text{AWSM}}(\theta_2)| > \frac{1}{3}\varepsilon)
$$
$$
< \mathbb{P}(|\hat{\mathcal{J}}_{\text{AWSM}}(\theta_1) - \mathcal{J}_{\text{AWSM}}(\theta_1)| + |\hat{\mathcal{J}}_{\text{AWSM}}(\theta_2) - \mathcal{J}_{\text{AWSM}}(\theta_2)| > \frac{1}{6}\varepsilon)
$$
$$
< \mathbb{P}(|\hat{\mathcal{J}}_{\text{AWSM}}(\theta_1) - \mathcal{J}_{\text{AWSM}}(\theta_1)| > \frac{1}{12}\varepsilon) \to 0, M \to \infty.
$$

Now we follow exactly the same steps as [2] for the uniform in probability convergence. Since Equation (24) hold, for such a $\delta$ in that equation, since $\Theta$ is a compact set in $\mathbb{R}^d$, there exists a finite number of open balls with radius $\delta$ whose union covers $\Theta$. Let $\vartheta_1, \ldots, \vartheta_i, \ldots, \vartheta_L$ denote the centers of these balls. We denote $\vartheta_{i(\theta)}$ the center of a ball which contains $\theta$. Since we have

$$
\mathbb{P}(\sup_\theta |\hat{\mathcal{J}}_{\text{AWSM}}(\theta) - \mathcal{J}_{\text{AWSM}}(\theta)| > \varepsilon) \le \mathbb{P}(\sup_\theta |\hat{\mathcal{J}}_{\text{AWSM}}(\theta) - \hat{\mathcal{J}}_{\text{AWSM}}(\vartheta_{i(\theta)})| > \frac{\varepsilon}{3})
$$
$$
+ \mathbb{P}(\sup_\theta |\hat{\mathcal{J}}_{\text{AWSM}}(\vartheta_{i(\theta)}) - \mathcal{J}_{\text{AWSM}}(\vartheta_{i(\theta)})| > \frac{\varepsilon}{3}) + \mathbb{P}(\sup_\theta |\mathcal{J}_{\text{AWSM}}(\theta) - \mathcal{J}_{\text{AWSM}}(\vartheta_{i(\theta)})| > \frac{\varepsilon}{3}).
$$

The third term on the right equals 0 because of its definition and the uniform continuous of $\mathcal{J}_{\text{AWSM}}(\theta)$. The first term converges to $0, M \to \infty$ by Equation (24). For the second term, we write

$$
\mathbb{P}(\sup_\theta |\hat{\mathcal{J}}_{\text{AWSM}}(\theta) - \mathcal{J}_{\text{AWSM}}(\vartheta_{i(\theta)})| > \frac{\varepsilon}{3}) < \sum_{i=1}^{L} \mathbb{P}(|\hat{\mathcal{J}}_{\text{AWSM}}(\theta_i) - \mathcal{J}_{\text{AWSM}}(\theta_i)| > \frac{\varepsilon}{3}) \to 0.
$$

Finally we obtain $\sup_{\theta \in \Theta} |\hat{\mathcal{J}}_{\text{AWSM}}(\theta) - \mathcal{J}_{\text{AWSM}}(\theta)| \xrightarrow{p} 0$, which completes the proof.

$\square$

## C.2  Proof of Theorem 5.4

First we restate the technical assumptions needed for the proof.

**Assumption 5.2.** *Assume there exists $\alpha > 1$ such that,*

$$\inf_{\theta : ||\theta - \theta^*|| \geq \delta} \mathcal{J}_{\mathbf{h}}(\theta) - \mathcal{J}_{\mathbf{h}}(\theta^*) \geq C_{\mathbf{h}} \delta^{\alpha}$$

*holds for any small $\delta$. Here, $C_{\mathbf{h}}$ is a positive constant that depends on the weight function $\mathbf{h}$ such that $C_{a\mathbf{h}} = aC_{\mathbf{h}}$ for any positive constant $a$.*

For this assumption, we assume that the optimal parameter $\theta^*$ is well-separated from other neighbouring parameters in terms of the population objective values. This is a standard assumption for Theorem 5.52 in [19]. This assumption will satisfy if $\nabla_{\theta}^2 \mathcal{J}_{\mathbf{h}}(\theta)$ is positive definite at $\theta^*$.

**Assumption 5.3.** *For $\forall n \in \mathbb{N}^+$, there exists $\dot{A}_n(\mathcal{T}), \dot{B}_n(\mathcal{T})$ such that,*

$$|A_n(\mathcal{T}, \theta_1) - A_n(\mathcal{T}, \theta_2)| \leq \dot{A}_n(\mathcal{T})||\theta_1 - \theta_2||, \quad |B_n(\mathcal{T}, \theta_1) - B_n(\mathcal{T}, \theta_2)| \leq \dot{B}_n(\mathcal{T})||\theta_1 - \theta_2||.$$

For this assumption, we assume the objective function has certain level of continuity. This assumption will realize if $\sup_{\theta \in \Theta} ||\nabla_{\theta} A_n(\mathcal{T}, \theta)||$ exisits for any $n$ and the same condition also applies to $B_n(\mathcal{T}, \theta)$.

We also define,

$$J_{\mathbf{h}}(\mathcal{T}; \theta) = \sum_{n=1}^{N_T} \left[ A_n(\mathcal{T}, \theta) h_n(\mathcal{T}) + B_n(\mathcal{T}, \theta) \frac{\partial h_n(\mathcal{T})}{\partial t_n} \right],$$

we see $\mathbb{E}[J_{\mathbf{h}}(\mathcal{T}; \theta)] = \mathcal{J}_h(\theta)$.

Now we begin the proof of Theorem 5.4.

*Proof.* First define

$$\dot{J}_{\mathbf{h}}(\mathcal{T}) = \sum_{n=1}^{N_T} \left[ \dot{A}_n(\mathcal{T}) h_n(\mathcal{T}) + \dot{B}_n(\mathcal{T}) \frac{\partial h_n(\mathcal{T})}{\partial t_n} \right].$$

Next we evaluate the bracketing number of such function class,

$$\mathcal{F}_{\delta} := \{ J_{\mathbf{h}}(\mathcal{T}; \theta) - J_{\mathbf{h}}(\mathcal{T}; \theta^*) | \theta \in \Theta, ||\theta - \theta^*|| \leq \delta \}.$$

a

Denote $\mathcal{N}_{[]}(\varepsilon, \mathcal{F}, L^2(P))$ as the bracketing number of $\mathcal{F}$ with the radius $\varepsilon$ under the norm of $L^2(P)$. $P$ is the underlying probability measure induced by $\mathcal{T}$. Use Example 19.6 in [19] we have,

$$\mathcal{N}_{[]}(\varepsilon ||J||_{L^2(P)}, \mathcal{F}_{\delta}, L^2(P)) \leq (1 + \frac{4\delta}{\varepsilon})^r$$

Since

$$|J_{\mathbf{h}}(\mathcal{T}; \theta) - J_{\mathbf{h}}(\mathcal{T}; \theta^*)| \leq \dot{J}_{\mathbf{h}}(\mathcal{T})||\theta - \theta^*|| \leq \dot{J}_{\mathbf{h}}(\mathcal{T})\delta, \tag{25}$$

therefore $\dot{J}_{\mathbf{h}}(\mathcal{T})\delta$ is the envelope of $\mathcal{F}_{\delta}$, we denote it as $F_{\delta}$. Therefore we obtain,

$$\mathcal{N}_{[]}(\varepsilon ||F_{\delta}||_{L^2(P)}, \mathcal{F}_{\delta}, L^2(P)) = \mathcal{N}_{[]}(\varepsilon \delta ||\dot{J}_{\mathbf{h}}||_{L^2(P)}, \mathcal{F}_{\delta}, L^2(P)) \leq (1 + \frac{4}{\varepsilon})^r.$$

After obtaining this quantity, the next step is to upper bound the empirical process defined as,

$$\mathbb{G}_M \big( J_{\mathbf{h}}(; \theta) - J_{\mathbf{h}}(; \theta^*) \big) := \frac{1}{\sqrt{M}} \sum_{m=1}^{M} \big[ J_{\mathbf{h}}(\mathcal{T}_m; \theta) - J_{\mathbf{h}}(\mathcal{T}_m; \theta^*) - \mathbb{E}_{p(\mathcal{T})}[J_{\mathbf{h}}(\mathcal{T}; \theta) - J_{\mathbf{h}}(\mathcal{T}; \theta^*)]\big]$$

Using Corollary 19.35 in [19], since we have $||\dot{J}_{\mathbf{h}}||_{L^2(P)} = \Gamma(\mathbf{h}, A, B) < \infty$, then,

$$
\begin{aligned}
\mathbb{E}\left[\sup_{f \in \mathcal{F}_\delta} \mathbb{G}_n(f)\right] &\leq C J_{[]}(||F||_{L^2(P)}, \mathcal{F}_\delta, L^2(P)) \\
&= C||F||_{L^2(P)} \int_0^1 \sqrt{\log N_{[]}(\varepsilon||F||_{L^2(P)}, \mathcal{F}_\delta, L^2(P))} d\varepsilon \quad (26) \\
&\leq C\delta||\dot{m}||_{L(P)} \sqrt{r} \int_0^1 \sqrt{\log(1 + \frac{4}{\varepsilon})} d\varepsilon \\
&\leq C'\sqrt{r}\delta\Gamma(\boldsymbol{h}, A, B),
\end{aligned}
$$

where $C$ and $C'$ are universal constant and $J_{[]}(\cdot, \cdot, \cdot)$ is the entropy integral.

Finally, given that $\hat{\theta}$ converges to $\theta^*$ in probability, combined with Assumption 5.2 and Equation (26), using Theorem 5.52 in [19] we have, for $\delta < CK_\alpha \frac{\sqrt{r}}{2^{\alpha-1}} \frac{\Gamma(\mathbf{h}, A, B)}{C_{\mathbf{h}}}$, we have,

$$
\Pr\left[||\hat{\theta} - \theta^*|| \leq \left(CK_\alpha \frac{\Gamma(\mathbf{h}, A, B)}{\delta C_{\mathbf{h}}} \sqrt{\frac{r}{n}}^{-1/(\alpha-1)}\right)\right] \geq 1 - \delta,
$$

where $C$ is a universal constant and $K_\alpha = \frac{2^{2\alpha}}{2^{\alpha-1}-1}$.

$\square$

### C.3 Proof of Theorem 5.5

First we give a rigorous definition of $\mathcal{H}$,

$$
\begin{aligned}
\mathcal{H} := \Big\{&\mathbf{h}(\mathcal{T}) \mid \mathbf{h} : \mathbb{R}_+^{N_T} \to \mathbb{R}_+^{N_T},\ h_n(\mathcal{T}) = h_n(t_n, \ldots, t_1), h_n(\mathcal{T})|_{t_n=t_{n-1}} = h_n(\mathcal{T})|_{t_n=T} = 0 \\
&|h_n(t_n^1, t_1, \ldots, t_{n-1}) - h_n(t_n^2, t_1, \ldots, t_{n-1})| \leq |t_n^1 - t_n^2|, \forall n, t_n^1, t_n^2\Big\}.
\end{aligned}
$$
(27)

In other words, $\mathcal{H}$ contains functions whose component function $h_n$ is 1-Lipschitz continuous w.r.t. its last dimension $t_n$. It's easy to verfity $\mathbf{h}^0 \in \mathcal{H}$.

*Proof.* First we prove that,
$$
\max_{\mathbf{h} \in \mathcal{H}} \mathcal{L}_{\mathbf{h}}(\theta) = \mathcal{L}_{\mathbf{h}^0}(\theta) \quad (28)
$$

Notice that,

$$
\mathcal{L}_{\mathbf{h}}(\theta) = \frac{1}{2}\mathbb{E}_{p(\mathcal{T})}\left[\sum_{n=1}^{N_T} (\psi(t_n|\mathcal{F}_{t_{n-1}}) - \psi_\theta(t_n|\mathcal{F}_{t_{n-1}}))^2 h_n(\mathcal{T})\right]
$$

Notice that, in $\mathcal{H}$, for any $z = t - n - 1$ or $z = T$,
$$
|h_n(\mathcal{T})| = |h_n(t_n, \ldots, t_1)| = |h_n(t_n, \ldots, t_1) - h_n(z, \ldots, t_1)| \leq |t_n^1 - z| = h_n^0(\mathcal{T})
$$

Therefore,

$$
\mathcal{L}_{\mathbf{h}}(\theta) \leq \frac{1}{2}\mathbb{E}_{p(\mathcal{T})}\left[\sum_{n=1}^{N_T} (\psi(t_n|\mathcal{F}_{t_{n-1}}) - \psi_\theta(t_n|\mathcal{F}_{t_{n-1}}))^2 h_n^0(\mathcal{T})\right] = \mathcal{L}_{\mathbf{h}^0}(\theta), \forall \mathbf{h} \in \mathcal{H}
$$

and equation 28 is proved.

Finally, we have,

$$
\begin{aligned}
&\inf_{\theta:||\theta-\theta^*||\geq\delta} \mathcal{J}_{\mathbf{h}^0}(\theta) - \mathcal{J}_{\mathbf{h}^0}(\theta^*) \\
=&\inf_{\theta:||\theta-\theta^*||\geq\delta} \mathcal{L}_{\mathbf{h}^0}(\theta) - \mathcal{L}_{\mathbf{h}^0}(\theta^*) \\
=&\inf_{\theta:||\theta-\theta^*||\geq\delta} \sup_{\mathbf{h}\in\mathcal{H}}[\mathcal{L}_{\mathbf{h}}(\theta) - \mathcal{L}_{\mathbf{h}}(\theta^*)] \\
=&\inf_{\theta:||\theta-\theta^*||\geq\delta} \sup_{\mathbf{h}\in\mathcal{H}}[\mathcal{J}_{\mathbf{h}}(\theta) - \mathcal{J}_{\mathbf{h}}(\theta^*)] \\
\geq&\sup_{h\in\mathcal{H}} \inf_{\theta:||\theta-\theta^*||\geq\delta}[\mathcal{J}_{\mathbf{h}}(\theta) - \mathcal{J}_{\mathbf{h}}(\theta^*)].
\end{aligned}
$$

The second equality is due to Equation (28) and $\mathcal{L}_{\mathbf{h}}(\theta^*) = 0$. The inequality is due to max-min inequality. $\qquad\square$

## C.4 Continued Discussion on Optimal Weight Function

Given $\boldsymbol{h}^0$ maximizes $C_{\boldsymbol{h}}$, it is still unclear whether it is still preferable considering $\Gamma(\mathbf{h}, A, B)$. This is indeed a tough question that we do not yet have a satisfying answer. For specific parametric models, $\dot{A}_n(\mathcal{T}), \dot{B}_n(\mathcal{T})$ can be computed analytically (see line 478-480) and then $\Gamma(\mathbf{h}, A, B)$ can be computed via Monte Carlo. Then we can study how sensitive $\Gamma(\mathbf{h}, A, B)$ is to $\mathbf{h}$. For general models, especially when $\psi_\theta$ is a deep neural network like THP or SAHP, $\Gamma(\mathbf{h}, A, B)$ is intractable to compute.

However, heuristically speaking, our choice of near-optimal weight function $\mathbf{h}^0$ should be a good choice even concerning $\Gamma(\mathbf{h}, A, B)$. To make $\Gamma(\mathbf{h}, A, B)$ small, a natural idea is to make $|h_n(\mathcal{T})|$ and $|\frac{\partial}{\partial t_n} h_n(\mathcal{T})|$ small. The weight function we chose and its derivative have relatively positive low powers with respect to $t_n$, therefore making $|h_n^0(\mathcal{T})|$ and $|\frac{\partial}{\partial t_n} h_n^0(\mathcal{T})|$ small. For weight functions $h_n^1(\mathcal{T}) = (T - t_n)(t_n - t_{n-1})$, the power w.r.t. $t_n$ is two. And for $h_n^2(\mathcal{T}) = \sqrt{(T - t_n)(t_n - t_{n-1})}$, its derivative is $\frac{\partial}{\partial t_n} h_n^2(\mathcal{T}) = \frac{T - t_n - (t_n - t_{n-1})}{\sqrt{(T - t_n)(t_n - t_{n-1})}}$, the numerator is usually a bounded quantity and the denominator may be close to zero, making its derivative large. In conclusion, $\mathbf{h}^0$ is a better choice compared with $\mathbf{h}^1$ or $\mathbf{h}^2$ concerning $\Gamma(\mathbf{h}, A, B)$.

## C.5 Discussion on Poisson Process

For Poisson process, results in Section 5 also holds, including the consistency and the convergence rate. For the choice of weight function, a reasonable choice is still the distance function presented below.

Consider a Poisson process, since the support of $\mathcal{T}$ is $V = \{\mathcal{T} \in N^T, 0 \le t_1 \le \ldots \le t_{N_T} \le T\}$. Define $A \in \mathbb{R}^{(N_T+1) \times N_T}$ to be a coefficient matrix used below as,

$$
A = \begin{bmatrix}
-1 & 0 & 0 & \ldots & 0 \\
1 & -1 & 0 & \ldots & 0 \\
0 & 1 & -1 & \ldots & 0 \\
\vdots & \vdots & \vdots & \ddots & 0 \\
0 & 0 & 0 & 1 & -1 \\
0 & 0 & 0 & 0 & 1
\end{bmatrix}.
$$

Denote $\mathbf{a}_n \in \mathbb{R}^{N_t}$ as the $n$-th row vector of $A$. Let $b_n = 0, n \in [N_T], b_{N_T+1} = -T$. Then $V$ is a convex Polytope which can be rewritten as,

$$
V = \{\mathcal{T} \in \mathbb{R}^{N_T} | \langle \mathbf{a}_n, \mathcal{T} \rangle + b_n \le 0, n = 1, 2, \ldots, N_T + 1\}.
$$

Therefore, the distance between $\mathcal{T}$ and $\partial V$ is ,

$$
dist(\mathcal{T}, \partial V) = \min_{\mathbf{z}} \{\|\mathcal{T} - \mathbf{z}\| \max_{n \in [N_T+1]} [\langle \mathbf{a}_n, \mathbf{z} \rangle + b_n] = 0\} = \min_{n \in [N_T+1]} \frac{|\langle \mathbf{a}_n, \mathbf{z} \rangle + b_n|}{\|\mathbf{a}_n\|}. \tag{29}
$$

Let $h_n^0(\mathcal{T}) = dist(\mathcal{T}, \partial V), \mathbf{h}^0(\mathcal{T}) = (h_n^0(\mathcal{T}), h_n^0(\mathcal{T}), \ldots, h_n^0(\mathcal{T}))^T$. Again, $\mathbf{h}^0(\mathcal{T})$ is only weak derivative. The envelope theorem in [14] yields,

$$
\nabla_{\mathcal{T}} \mathbf{h}^0(\mathcal{T}) = -\frac{\mathbf{a}_{n^*}}{\|\mathbf{a}_{n^*}\|},
$$

wehre $n^*$ is the is the minimizer of the last minimization in Equation (29).

Similar arguments as in Section 5.3 also applies to Poisson process, with the distance weight function defined above. Such a weight function also maximizes the denominator for the convergence rate of the Poisson process. However, in the experiment, we adtops another weight function for easier implementation defined as $\mathbf{h}^1(\mathcal{T})$ with its $n$-th component $h_n^1(\mathcal{T}) = (T - t_n)(t_n - t_{n-1})$.

Table 3: The MAE of 2-variate Hawkes processes trained by MLE, (A)SM, and (A)WSM on the synthetic dataset.

| Estimator | Exp-Hawkes | | | Gaussian-Hawkes | | | |
|---|---|---|---|---|---|---|---|
| | $\alpha_{21}$ | $\alpha_{22}$ | $\mu_2$ | $\alpha_{12}$ | $\alpha_{21}$ | $\alpha_{22}$ | $\mu_2$ |
| (A)WSM | $\mathbf{0.052_{\pm 0.054}}$ | $\mathbf{0.022_{\pm 0.005}}$ | $\mathbf{0.011_{\pm 0.012}}$ | $0.037_{\pm 0.043}$ | $0.082_{\pm 0.080}$ | $0.012_{\pm 0.010}$ | $0.060_{\pm 0.066}$ |
| (A)SM | $0.769_{\pm 0.001}$ | $0.769_{\pm 0.001}$ | $0.680_{\pm 0.270}$ | $0.126_{\pm 0.108}$ | $0.971_{\pm 0.040}$ | $0.717_{\pm 2.85}$ | $2.507_{\pm 1.957}$ |
| MLE | $0.065_{\pm 0.032}$ | $0.034_{\pm 0.015}$ | $\mathbf{0.014_{\pm 0.002}}$ | $\mathbf{0.025_{\pm 0.032}}$ | $\mathbf{0.045_{\pm 0.041}}$ | $\mathbf{0.006_{\pm 1.06}}$ | $\mathbf{0.051 \pm 0.049}$ |

Table 4: The hyparameters for experiments in Table 2.

| Dataset | Epochs | | $\alpha_{\text{AWSM}}$ | | Trunc | | $\alpha_{\text{DSM}}$ | | $\sigma_{\text{DSM}}$ | |
|---|---|---|---|---|---|---|---|---|---|---|
| | SAHP | THP | SAHP | THP | SAHP | THP | SAHP | THP | SAHP | THP |
| Half-Sin | 100 | 100 | 20 | 20 | F | F | 20 | 10 | 0.01 | 0.01 |
| Stackoverflow | 100 | 500 | 20 | 20 | T | F | 10 | 10 | 0.1 | 0.1 |
| Taobao | 500 | 300 | 50 | 20 | T | T | 20 | 20 | 0.01 | 5e-3 |
| Retweets | 100 | 100 | 20 | 10 | F | F | 2000 | 10 | 0.01 | 5e-4 |

# D  Additional Experimental Details

In this section, we present some experimental details. First we discuss our modification to the original AWSM for when $T$ for a dataset is not accessible. Then we provide addtional estimation results for the parametric model and hyperparameters for results in Table 2.

## D.1  Weight for Equal Length Sequences

In this paper, we consider the setting of the temporal point process in a fixed time interval $[0, T]$ consistently. Technically speaking, this poses extra difficulty since the dimension(number of events) would also be random. Therefore, the realized point processes should have a random length in this setting, which is the case for our dataset. However, in real data collection, different sequences may be sampled from different lengths of time interval lengths, and our proposed weight may fail under such a case, resulting in inconsistent estimation. In such a case, we propose to truncate the sequences to be of same length, and introduce the weight function for fixed-length temporal point processes.

We focus on AWSM for Hawkes process. A fixed $N$-length Hawkes process could be considered as a $N$ dimensional random variable with density function as,

$$p(t_1, \ldots, t_N) = \prod_{n=1}^{N} \lambda^*(t_n) \int_0^{t_N} \lambda^*(t) dt.$$

Since the number of observations is fixed, we could simply treat it as an $N$ dimensional random variable and perform autoregressive weighted score matching. The valid weight function in this case should satisfy $h_n(\mathcal{T})|_{t_n=t_{n-1}} = h_n(\mathcal{T})|_{t_n=t_{n+1}} = 0$. Similar nonasymptotic bounds could be derived here and a near optimal weight function would be the Euclidean distance between $t_n$ and the boundary of $t_n$'s support, which would be $\{t_{n-1}, t_{n+1}\}$. Therefore, the near-optimal weight function would be $h_n^0(\mathcal{T}) = \min\{t_n - t_{n-1}, t_{n+1} - t_n\}$.

During experiments, when $T$ is unknown and cannot be approximated well, we perform a data truncation for each batch. We naively drop all timestamps beyond the length of the shortest sequence to make length consistent within a batch. Then we perform an AWSM for the fixed-length Hawkes process. We specify the results with this modification when we show our hyperparameters.

## D.2  Addtional Results and Hyperparameters

For parametric models, Table 3 provide the estimation results for some parameters and Figure 3 shows the learned intensity functions on the 2-nd dimension for the 2-variate Hawkes processes.

For deep point process experiments, we run 4 datasets with 3 methods deployed on 2 models. We show the hyperparameters of those experiments in Table 4. **Epochs** column represents the number

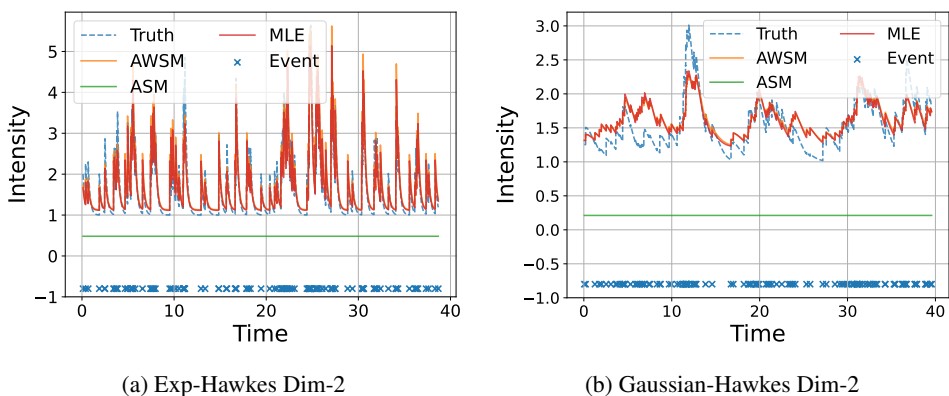

(a) Exp-Hawkes Dim-2                     (b) Gaussian-Hawkes Dim-2

Figure 3: The learned intensity functions from MLE, (A)SM, and (A)WSM on (a) Exp-Hawkes and (b) Gaussian-Hawkes for the 2-nd dimension.

of epochs for the experiments of a model (THP or SAHP) on a dataset. We train same epochs for three training methods (MLE, AWSM or DSM) and validate every 10 epochs to report the best result. When training MLE, the hyperparameter is the number of integral nodes, which is always 10 for all experiments. When training AWSM, we have two hyperparameters, the value of balancing coefficient for CE loss, shown in column $\alpha_{\text{AWSM}}$ and whether data truncation is performed as shown in column **Trunc**. T represents performing data truncation and F represents no data truncation. For DSM, the hyperparameters are balancing coefficient denoted as $\alpha_{\text{DSM}}$ and variance of noise denoted as $\sigma_{\text{DSM}}$.

