# OpenReview forum: "Is Score Matching Suitable for Estimating Point Processes?"
_NeurIPS.cc/2024/Conference — NeurIPS 2024 poster_

### Official Review · Reviewer_X8YN · 2024-07-04

**Soundness:** 3
**Presentation:** 3
**Contribution:** 3
**Rating:** 7
**Confidence:** 4

**Summary:**

EDIT: I have changed my score from a "weak accept" to an "accept."

The paper studies the potential use of score matching (SM) to do inference
with the model of interest is a Poisson point process or Hawkes
processes. Because using SM in practice requires modifying the objective
with integration by parts, the paper shows for many common Poisson and
Hawkes processes a core assumption to make this modification possbile does
not hold. In order to remedy the situation, the paper proposes a random weighted
version of SM called WSM (along with an analog for Autoregressive Score
Matching (ASM) called WASM), which reestablishes the validity of integration
by parts for the aforementioned processes. The paper argues theoretically
that their new estimators are consistent, and it provides empirical
analyses to demonstrate that the new estimator outperforms the non-weighted
variant and matches the performance of alternatives. It also makes an
argument to select a weighting function based on some set of assumptions.

**Strengths:**

The paper points out a fundamental issue with trying to use SM for general
Poisson point processes and Hawkes processes, and in addition, it recommends
a weighting scheme that patches up this problem based on some reasonable
assumptions. This is a valuable observation, and the proposed remedy looks
reasonable. There is ample empirical work to suggest that their tweak on SM
and ASM does agree with the maximum likelihood estimator (MLE). The theory
provided in the paper also argues that their weighted estimator is
consistent if the model is well-specified.

Originality:

[+]

+ The authors have applied a weighting scheme to SM and ASM that permits them
to use integration by parts for the standard objective and prove the
consistency of their new estimator under suitable regularity conditions.

Quality:

[+]

+ The empirical work is thorough and does a good job checking the accuracy
  of their new estimator.

Clarity:

[+]

+ The presentation of the theory and empirical work is very neat and
  polished.
+ Assumptions are clearly laid out to ensure consistency of the new estimator.

Significance:

[+]

+ The new estimator provides the ability to do consistent semiparametric
  estimation for Poisson point and Hawkes processes with SM (instead of
  MLE), which provides a new tool for very large dimensional problems.

**Weaknesses:**

There are a couple critiques of the paper:

- While the idea of using a weighting function to enable integration by
  parts is a good one, it has been seen in other fields before
  (e.g. weighting Stein kernels in compact settings), and hence is not
  completely novel.
- The argument in Section 5 outlining a weighting function is a great
  addition to the paper, but given that it is only optimizing $C_h$ and not
  $\Gamma(h, A, B)$, it is hard to know how close it is to the optimum.

Originality:

[-]

- The only novel insight of the paper is that the trick of integration by parts,
  so often using in score matching, is not possible for general point
  processes, but it can be recovered by weighting the Fisher divergence a bit
  differently. It is not clear if this idea alone is sufficient for publication.

Quality:

[-]

- The argument to choose their weighting function $h$ is definitely better
  than simply picking one. That being said, it still not clear how close to
  optimal their choice of $h$ is given they are only optimizing one term in
  the objective ($C_h$). The paper would be a bit stronger if they could
  include a different (natural) weighting function to show their new choice
  is an improvement.

Clarity:

[-]

- Equation 1: Can we at least have a sentence explaining the paper will be
  slightly abusing notation by referring to $p(T)$ as the conditional
  distribution of $p$ given $N_t = |T|$?

**Questions:**

[Q1] Given the choice of $h$ optimizes $C_h$ but ignores $\Gamma(h, A, B)$,
how sensitive is $\Gamma$ to the choice of $h$? This could be demonstrated
empirically or theoretically, but it would be good to have some analysis of
this to argue why this weighting function is decent.

---

> ### Author Rebuttal · Authors · 2024-08-07
>
> Thank you so much for reviewing our paper. We answer your questions below.
> > The argument to choose their weighting function $h$ is definitely better than simply picking one. That being said, it still not clear how close to optimal their choice of h is given they are only optimizing one term in the objective $C_{\bf h}$. The paper would be a bit stronger if they could include a different (natural) weighting function to show their new choice is an improvement.
>
> We thank the reviewer for pointing this out. To address your concern, we carry out experiments and compare three different weight functions that satisfy the criterion (Eq. 15) for AWSM on Hawkes process. Other than the near-optimal weight function $\bf{h}^0$ introduced in our paper, we also consider a natural weight function $\bf{h}^1$ defined as,
>
> $
> h_n^1(t_n)=(t_n-t_{n-1})(T-t_{n}).
> $
>
> We also consider another valid weight function: the square root of $\bf h^1$ denoted as $\bf h^2$,
>
> $
> h_n^2(t_n)=\sqrt{(t_n-t_{n-1})(T-t_n)}.
> $
>
> All three weight functions can be applied in AWSM to recover ground-truth parameters, however with different convergence rates. We carry out experiments on synthetic data for exponential-decay model with the same setting as section 6.2 in our paper. We measure their MAE for different sample sizes in Figure 1 of the attached PDF and find that $\bf h^0$ does achieve the best results among the three weight functions.
>
> We hope these additional experimental results can enhance the credibility of our paper and address your concerns. We would definitely add these results in the camera ready.
>
>
> > Given the choice of $\bf h$ optimizes $C_{\bf h}$ but ignores $\gamma(\bf h,A,B)$, how sensitive is $\Gamma$ to the choice of $\bf h$? This could be demonstrated empirically or theoretically, but it would be good to have some analysis of this to argue why this weighting function is decent.
>
> We thank the reviewer for pointing this out. This is indeed a tough question that we do not yet have a satisfying answer. For specific parametric models, $\dot A_n(\mathcal T), \dot B_n(\mathcal T)$ can be computed analytically (see line 478-480) and then $\Gamma(\mathbf{h},A,B)$ can be computed via Monte Carlo. Then we can study how sensitive $\Gamma(\mathbf{h},A,B)$ is to $\bf h$. For general models, especially when $\psi_{\theta}$ is a deep neural network like THP or SAHP, $\Gamma(\mathbf{h},A,B)$ is intractable to compute.
>
> However, heuristically speaking, our choice of near-optimal weight function $\bf h^0$ should be a good choice even concerning $\Gamma(\mathbf{h},A,B)$. To make $\Gamma(\mathbf{h},A,B)$ small, a natural idea is to make $|h_n(\mathcal T)|$ and $|\frac{\partial}{\partial t_n}h_n(\mathcal T)|$ small. The weight function we chose and its derivative have relatively positive low powers with respect to $t_n$, therefore making $|h_n^0(\mathcal T)|$ and $|\frac{\partial}{\partial t_n}h_n^0(\mathcal T)|$ small. For  weight functions  $h_n^1(\mathcal T)=(T-t_n)(t_n-t_{n-1})$, the power w.r.t. $t_n$ is two. And for $h^2_n(\mathcal T)=\sqrt{(T-t_n)(t_n-t_{n-1})}$, its derivative is $\frac{\partial}{\partial t_n}h_n^2(\mathcal T)=\frac{1}{2}\frac{T-t_n-(t_n-t_{n-1})}{\sqrt{(T-t_n)(t_n-t_{n-1})}}$ , the numerator is usually a bounded quantity and the denominator may be close to zero, making it large. In conclusion, $\bf h^0$ is a better choice compared with $\bf h^1$ or $\bf h^2$ concerning $\Gamma(\mathbf{h},A,B)$.
>
>
>
>
> > Equation 1: Can we at least have a sentence explaining the paper will be slightly abusing notation by referring to $p(\mathcal T)$ as the conditional distribution of p given $N_T=|T|$?
>
> We thank the reviewer for pointing this out.  Here since $N_T$ is random, by the notation $p(\mathcal T)$, we are referring to the probability density or the likelihood of observing $N_T$ events $t_1,\ldots, t_{N_T}$  in $[0,T]$.  For Poisson process, the conditional distribution of event time $t_1,\ldots, t_{N}$ given $N_T = N$ is,
>
> $
> p(t_1,\ldots, t_N|N_T = N) = \frac{\prod_{n=1}^N \lambda(t_n)}{\int_{0\leq t_1\dots\leq t_N\leq T}\prod_{n=1}^{N}\lambda(t_n)dt_1\ldots dt_N}, 0\leq t_1\ldots\leq t_{N}\leq T.
> $
>
> We acknowledge that our paper may not employ the rigorous notation typically used in the fields of probability and stochastic processes. Instead, we have opted for notations that we believe are accessible to readers from both the computer science and statistics communities. We deeply apologize for any confusion this may have caused.

---

> > ### Comment · Reviewer_X8YN · 2024-08-09
> >
> > Thanks for your reply. I appreciate that you have done some extra work to demonstrate the choice of $h^0$ is better than other obvious alternatives. I am inclined to accept this paper.

---

> > > ### Author Response · Authors · 2024-08-09
> > > **Thanks**
> > >
> > > Thank you very much for your recognition and support. We will include the additional comparison of different weight functions in the camera-ready version. Thank you once again for your constructive feedback and for increasing your rating to accept.

---

### Official Review · Reviewer_B3db · 2024-07-13

**Soundness:** 3
**Presentation:** 3
**Contribution:** 2
**Rating:** 5
**Confidence:** 4

**Summary:**

This paper studies the use of score matching for estimation in point process models, and proves the incompleteness in the original score matching estimators due to the bounded support. Weighted score matching is use to address the issue and theoretical results are establish ed for the consistency and optimality of the proposed estimator. Experiments on different data demonstrate the effectiveness of the proposed estimator which yields results comparable to maximum likelihood estimation.

**Strengths:**

1. This paper theoretically shows the limitation in the use of original score matching estimators for point processes, pointing out their incompleteness and provide a solution.
2. Theoretical analysis is given to support the claim and the convergence of the proposed method.
3. The paper is well-written and easy to follow.

**Weaknesses:**

1. As acknowledged in the limitation section in the paper, existing approaches that adopt denoising score matching on the point process is not considered. Even with theoretical gaurantee, experimentally, such baselines would be necessary to include in order to fully demonstrate the superiority of the method.
2. For Hawkes processes, the problem cause by the bounded support can be solved by applying log-normalization to transform the bounded temporal domain into an unbounded one [1]. One could achieve this by removing the right bound and transform the $(0,+\infty)$ to $(-\infty,+\infty)$. Although the real data is observed within finite window, the effect of removing the right bound should be negligible as it only affect the last event.
3. Including more metrics on event prediction, such as MSE of event time, would be helpful, as the practical use of point process model is for prediction.

[1] Lin, H., Wu, L., Zhao, G., Liu, P., & Li, S. (2022). Exploring generative neural temporal point process. Transactions on Machine Learning Research.

**Questions:**

* Did the author observe any instability during the score matching training?

**Limitations:**

Included in weakness

---

> ### Author Rebuttal · Authors · 2024-08-07
>
> Thank you so much for reviewing our paper. We answer your questions below.
> > Denoising score matching for point processes is not considered.
>
> We thank the reviewer for pointing this out. Indeed, denoising score matching (DSM) is not considered in our paper because we mainly focus on correcting the original score matching. However, to address your concerns, we have carried out experiments on DSM. **We can conclude that DSM is inferior to AWSM in terms of both efficiency and accuracy.**
>
> We deploy DSM on THP and SAHP. We use the DSM loss as in [1]. For observed timestamps $t_n^{(m)}$ in $m$-th sequence, we sample $L$ noise samples $\tilde t_{n,l}^{(m)}=t_n^{(m)}+ \epsilon_{n,l}^{(m)}, l = 1,\ldots, L,$ where $\text{Var}(\varepsilon_{n,L}^{(m)})=\sigma^2$ and get the DSM objective:
>
> $
> \hat {\mathcal J}(\theta)=\frac{1}{M}\sum_{m=1}^M\sum_{n=1}^{N_m}\sum_{l=1}^{L}\frac{1}{2L}[\psi_\theta(\tilde t_{n,l}^{(m)})+\frac{\varepsilon_{n,l}^{(m)}}{\sigma^2}]+\alpha\hat {\mathcal J}_\text{CE}(\theta),
> $
>
> where $\hat {\mathcal J}_{\text{CE}}(\theta)$ is the cross-entropy loss as line 191-192 in our paper. We carried out experiments on synthetic data where $T$ is known. We compared the TLL, ACC, and running time of DSM with MLE and AWSM. The comparison results are shown in Table 1 in the attached PDF. DSM performs the worst among the three estimators, and AWSM is the fastest.
>
> The reasons why DSM performs poorly are as follows:
>
> 1. In terms of accuracy, as discussed in Section 5 of [2], DSM is biased when $\sigma > 0$, while our AWSM is unbiased and produces consistent estimates. Therefore, it is not surprising to see that AWSM achieves better TLL and ACC than DSM.
>
> 2. In terms of efficiency, previous work states that DSM is faster than SM because the Hessian matrix is expensive to compute, while DSM avoids this. However, as presented in [3], ASM (or AWSM) also avoids computing the Hessian matrix by replacing it with a one-dimensional partial derivative  $\frac{\partial}{\partial t_n}\psi_{\theta}(t_n|\mathcal F_{t_{n-1}})$. Therefore, DSM is not more efficient than AWSM for avoiding the computation of the Hessian matrix. On the contrary, DSM can be slower than AWSM because DSM requires $L$ noise samples for each timestamp and must compute their scores.
>
>
>
> > For Hawkes processes, the problem can be solved by applying log-normalization to transform the bounded temporal domain into an unbounded one [1].
>
> This is a misunderstanding. **Apply log-normalization does not provide any solution to the problem, we can prove that the integration by parts still produces an intractable term after taking this transformation. Therefore, using ASM on a log-transformed sequence still results in wrong estimation**. If one hopes to get consistent estimation after log transformation, a suitable weight function is still needed. This is summarised as follows.
>
> |                                                      | ASM  | AWSM |
> | ---------------------------------------------------- | ---- | ---- |
> | No transformation                                    | ×    | √    |
> | Log Transform $x_n = \log t_n$                       | ×    | √    |
> | Log Transform on interval $y_n = \log (t_n-t_{n-1})$ | ×    | √    |
>
> We illustrate this using the simplest example both theoretically and empirically. Consider a homogeneous Poisson process with intensity $\lambda ^*$ with observations $t_1, \ldots, t_{N_T}$ over $0\leq t_1\leq \ldots \leq t_{N_T}\leq T$. Now we consider
>
> 1. First apply log transformation to timestamps then using ASM. The estimate is denoted as $\hat \lambda_{\text{ASM,Log}}$.
>
> 2. First apply log transformation to intervals then using ASM on intervals. The estimate is denoted as $\hat \lambda_{\text{ASM,Log Interval}}$.
>
> For log-transformation on timestamp, we have $x_n=\log t_n$.
> Therefore the conditional pdf of $x_n$ given $\mathcal F_{x_{n-1}}$ is:
>
> $
> p(x_n|\mathcal F_{x_{n-1}})=\lambda \exp\left[-\lambda \exp(x_n)\right]\exp(x_n),
> $
>
> and we can compute conditional score accordingly. We plug these terms into the implicit ASM objective and get the estimate. Suppose we observed $M$ sequences of $t_1^{(m)}, \ldots, t_{N_m}^{(m)}$, after derivation we get an analytical form of $\hat \lambda_{\text{ASM,Log}}$:
>
> $
> \hat \lambda_{\text{ASM,Log}} =2\frac{\sum_{m=1}^M\sum_{n=1}^{N_m}\exp(x_n)}{\sum_{m=1}^M\sum_{n=1}^{N_m}\exp(2x_n)}.
> $
>
> Similarly, we consider transformation on intervals $\tau_n:= t_n-t_{n-1}, n\geq 2, \tau_1:=t_1, y_n = \log (\tau_n)$ and consider the score $\psi(y_n|\mathcal F_{y_{n-1}})$ then apply ASM. The estimate in this case is:
>
> $
> \hat \lambda_{\text{ASM,Log Interval}} =2\frac{\sum_{m=1}^M\sum_{n=1}^{N_m}\exp (y_n)}{\sum_{m=1}^M\sum_{n=1}^{N_m}\exp(2y_n)}.
> $
>
> Both estimates are wrong and can never recover the true parameter regardless of sample size. To compare, after transformation, suitable weight functions can be added. For log transformation on timestamps, we use weight function $h_n(x_n)=(x_n-x_{n-1})(\log T-x_n)$. For log transformation on intervals, we use $h_n(y_n) = \log [T-t_{n-1}]-y_n$. The corresponding estimates are $\hat \lambda_{\text{AWSM,Log}}$ and $\hat \lambda_{\text{AWSM,Log Interval}}$.
>
> We measure the MSE of these four estimates. Results are shown in Table 2 in the attached PDF. It is easy to see that as sample size increases, MSE of $\hat \lambda_{\text{ASM,Log}}$ and $\hat \lambda_{\text{ASM,Log Interval}}$ remains large and unchanged, showing that these two estimators are wrong regardless of sample size.
>
>
>
> > Including more metrics would be helpful.
>
> We will consider adding MSE to our revised paper.
>
> > Did the author observe any instability during training?
>
> For our weighted score matching, we did not notice instability.
>
> [1] SMURF-THP: score matching-based uncertainty quantification for transformer Hawkes process
>
> [2] A connection between score matching and denoising autoencoders
>
> [3] Autoregressive score matching

---

> > ### Comment · Reviewer_B3db · 2024-08-12
> >
> > Thank you sincerely for the additional experiments on denoising score matching and clarifications! For the log-normalization, could you kindly provide the detailed derivation for obtaining $\hat{\lambda}\_{\text{ASM,Log}}$ and $\hat{\lambda}\_{\text{ASM,Log Interval}}$? Additionally, is it possible to express $\hat{\lambda}\_{\text{AWSM,Log}}$ and $\hat{\lambda}\_{\text{AWSM,Log Interval}}$ in closed form?

---

> ### Author Response · Authors · 2024-08-12
>
> We greatly thank the reviewer for your constructive feedback.
>
> >  For the log-normalization, could you kindly provide the detailed derivation for obtaining $\hat \lambda_{\text{ASM,Log}}$ and $\hat \lambda _{\text{ASM,Log Interval}}$?
>
> Sure, we are more than delighted to discuss this with you.
>
> For log-normalization on timestamps, since the transformed variable $x_n$ has conditional pdf,
>
> $
> p(x_n|\mathcal F_{x_{n-1}})=\lambda \exp\left[-\lambda \exp(x_n)\right]\exp(x_n).
> $
>
> The conditional score and its partial derivative w.r.t. $x_n$ will be,
>
> $
> \psi(x_n|\mathcal F_{x_{n-1}}) = -\lambda \exp(x_n) + 1,
> $
>
> $
> \frac{\partial}{\partial x_n}\psi_{\theta}(x_n|\mathcal F_{x_{n-1}})=-\lambda \exp(x_n).
> $
>
> Now we perform ASM on the transformed variable $x_n$. Denote $\mathcal X = (x_1,\ldots, x_{N_T})^T$ as the vector of transformed timestamps,  the implicit ASM objective (Eq. 13) for $\mathcal X$ will be,
>
> $
> \mathcal J_\text{ASM}(\theta)=\mathbb E_{p(\mathcal X)}[\sum_{n=1}^{N_T}\frac{1}{2}\psi^2_\theta(x_n|\mathcal F_{x_{n-1}})+\frac{\partial}{\partial x_n}\psi_\theta(x_n|\mathcal F_{x_{n-1}})].
> $
>
> We plug in the conditional score we just computed into the above equation and get,
>
> $
> \mathcal J_{\text{ASM}}(\theta) =\mathbb E_{p(\mathcal X)}[\sum_{n=1}^{N_T}\frac{1}{2}\exp(2x_n)\lambda^2-\sum_{n=1}^{N_T}2\exp(x_n)\lambda ]+C,
> $
>
> where $C$ is a constant not containing $\theta$.
>
> The empirical objective based on samples $t_1^{(m)},\ldots t_{N_m}^{(m)}, m=1,\ldots M$ will be,
>
> $\hat J_\text{ASM}(\theta)=\frac{1}{M}\sum_{m=1}^M\sum_{n=1}^{N_m}[\frac{1}{2}\exp(2x_n^{(m)})\lambda^2-2\exp(x_n^{(m)})\lambda].
> $
>
>
> This is quadratic with respect to $\lambda$, and its minimal $\hat \lambda_{\text{ASM,Log}}$ can be computed analytically as
> $
> \hat \lambda_{\text{ASM,Log}} = 2\frac{\sum_{m=1}^M\sum_{n=1}^{N_m}\exp(x_n^{(m)})}{\sum_{m=1}^M\sum_{n=1}^{N_m}\exp(2x_n^{(m)})}.
> $
>
> For log transformation on time interval $y_n$, since the conditional pdf of $\tau_n = t_n-t_{n-1}$ is,
>
> $
> p(\tau_n|\mathcal F_{\tau_{n-1}})=\lambda \exp(-\lambda \tau_n),
> $
>
> thus the conditional pdf of $y_n = \log \tau_n$ will be,
>
> $
> p(y_n|\mathcal F_{y_{n-1}})=\lambda \exp\left[-\lambda \exp(y_n)\right]\exp(y_n),
> $
>
> it has the same analytical expression as that of $x_n$, except that $y_n$ and $x_n$ have different support. Therefore, following exactly the same steps as those of $x_n$, we can get the analytical expression of $\hat \lambda_{\text{ASM,Log, Interval}}$ in our rebuttal (with minor modification for adding superscript $(m)$).
>
> >  Additionally, is it possible to express $\hat \lambda_{\text{AWSM,Log}}$ and $\hat \lambda_{\text{AWSM, Log Interval}}$  in closed form?
>
> Yes, it is possible. We take $\hat \lambda_{\text{AWSM, Log}}$ as an example. The derivation of $\hat \lambda_{\text{AWSM, Log Interval}}$ is basically the same.
>
> First, the implicit  AWSM objective as Eq. 17 in our paper for $\mathcal X$ will be,
>
>
> $
> \mathcal J_\text{AWSM}(\theta)=\mathbb E_{p(\mathcal X)}[\sum_{n=1}^{N_T}\frac{1}{2}\psi^2_\theta(x_n|\mathcal F_{x_{n-1}})h_n(\mathcal X)+\frac{\partial}{\partial x_n}\psi_\theta(x_n|\mathcal F_{x_{n-1}})h_n(\mathcal X)+\psi_\theta(x_n|\mathcal F_{x_{n-1}})\frac{\partial}{\partial x_n}h_n(\mathcal X)].
> $
>
>
> Plug in $\psi_\theta(x_n|\mathcal F_{x_{n-1}})$ and $\frac{\partial}{\partial x_n}\psi_\theta(x_n|\mathcal F_{x_{n-1}})$ into the empirical version of the above equation and  get,
>
> $
> \hat J_\text{AWSM}(\theta)=\frac{1}{M}\sum_{m=1}^M\sum_{n=1}^{N_m}[\frac{1}{2}\exp(2x_n^{(m)})h_n(\mathcal X^{(m)})\lambda^2-2\exp(x_n^{(m)})h_n(\mathcal X^{(m)})\lambda - \exp(x_n^{(m)})\frac{\partial}{\partial x_n}h_n(\mathcal X^{(m)})\lambda].
> $
>
> This is still quadratic w.r.t $\lambda$, and its minimal will be,
>
> $
> \hat \lambda_{\text{AWSM, Log}}=\frac{\sum_{m=1}^M\sum_{n=1}^{N_m}[2\exp(x_n^{(m)})h_n(\mathcal X^{(m)})+\exp(x_n^{(m)})\frac{\partial}{\partial x_n}h_n(\mathcal X^{(m)})]}{\sum_{m=1}^M\sum_{n=1}^{N_m}\exp(2x_n^{(m)})h_n(\mathcal X^{(m)})}.
> $
>
> Since we choose weight $h_n(\mathcal X)=(x_n-x_{n-1})(\log T-x_n)$ with derivative $\frac{\partial}{\partial x_n}h_n(\mathcal X)=\log T - 2x_n+x_{n-1}$. Plug them in the above equation, then we get a closed form of $\hat \lambda_{\text{AWSM, Log}}  $.

---

> > ### Comment · Reviewer_B3db · 2024-08-12
> >
> > Thank you for the further derivation and clarification! For the log interval case, each $\tau_i$ should follow the same exponential distribution, so $\hat{\lambda}_{\text{ASM,Log Interval}}$ should approximate the correct $\lambda$ with a large sample size. Could you please clarify why this is considered incorrect? I might have misunderstood something.

---

> ### Author Response · Authors · 2024-08-12
>
> We thank greatly for the reviewer for further engaging in the discussion.
> > For the log interval case, each $\tau_i$ should follow the same exponential distribution, so $\hat{\lambda}_{\text{ASM,Log Interval}}$ should approximate the correct $\lambda$ with a large sample size.
>
> This is not true. It is widely known that for an exponential distribution on $[0,\infty)$, it indeed has expectation $\frac{1}{\lambda}$ and second moments $\frac{2}{\lambda^2}$. Then the average of exponential distribution should be equal to $\frac{1}{\lambda}$ when sample is large. However, this is not the case when considering the time interval in a Poisson process on $[0,T]$.
>
> Considering the numerator of $\hat \lambda_{\text{AWSM,Log Interval}}$, we **cannot** draw the conclusion that,
>
> $
> \lim_{M\rightarrow \infty}\mathbb E_{p}[\frac{\sum_{m=1}^M\sum_{n=1}^{N_m}\tau_n^{(m)}}{\sum_{m=1}^MN_m}]=\frac{1}{\lambda}
> $
>
> We cannot use CLT or other things here since $\sum_{m=1}^M N_m$ is random and each $\tau_n^{(m)}$ does not necessarily have the same marginal distribution since their support is different.
>
> To illustrate this, we compute $\frac{\sum_{m=1}^M\sum_{n=1}^{N_m}\tau_n^{(m)}}{\sum_{m=1}^MN_m}$ for $M=10^{6}, T = 1$ under different $\lambda^{*}$ and show that the expectation of it is not equal to $\frac{1}{\lambda}$.
>
> |                                                              | $\lambda=1$ | $\lambda = 2$ | $\lambda = 4$ | $\lambda = 8$ |
> | ------------------------------------------------------------ | ----------- | ------------- | ------------- | ------------- |
> | $\frac{\sum_{m=1}^M\sum_{n=1}^{N_m}\tau_n^{(m)}}{\sum_{m=1}^MN_m}$ | 0.3693      | 0.2836        | 0.1886        | 0.1094        |
>
> Also for the denominator, its expectation is not equal or asymptotically equal to $\frac{2}{\lambda^2}$ when taking average. Therefore, $\hat{\lambda}_{\text{ASM,Log Interval}}$ does not approximate the correct $\lambda$ with a large sample size.
>
> We give a brief explanation as follows. For a stochastic process in a finite window $[0,T]$. Each $\tau_i$ has support $[0, T-\sum_{i\leq n}\tau_i]$ instead of $[0,\infty)$. Therefore, firstly, each $\tau_i$ has different support and does not follow the same distribution. Secondly, for each $\tau_i$ , it has support different from that of the exponential distribution, which is $[0, \infty)$.
>
> In conclusion, when discussing a point process on a finite time window, the right bound's effect must be considered. And the case is different from sampling a fixed length time sequence over an infinite time window.

---

> ### Comment · Reviewer_B3db · 2024-08-13
>
> Thanks so much for providing concrete answers to my queries and issues! Most of my concerns have been resolved and I have increased my score. I believe it would be valuable to include a discussion on denoising score matching, as well as some recent related work on score matching with point processes [1,2].
>
> [1] Beyond Point Prediction: Score Matching-based Pseudolikelihood Estimation of Neural Marked Spatio-Temporal Point Process.
>
> [2] Exploring generative neural temporal point process.

---

> > ### Author Response · Authors · 2024-08-13
> > **Thanks**
> >
> > Thank you for your constructive feedback and for increasing your rating to accept.

---

### Official Review · Reviewer_Se8E · 2024-07-15

**Soundness:** 3
**Presentation:** 3
**Contribution:** 3
**Rating:** 6
**Confidence:** 4

**Summary:**

The paper considers the problem of utilizing score-matching approaches for point processes. The main motivations for the paper are the Poisson and Hawkes processes. Both of these model the repeated occurrences of events over a finite time interval $[0, T]$. The Poisson process models the occurrence of an event within a fixed infinitesimal interval with a fixed homogenous rate $\lambda$ while a Hawkes process is a self-excitation process where past events increase the likelihood of future ones. Score-matching is a natural approach in settings where the computation of the normalizing constant of a parameterized model are challenging to compute. However, to ensure traceability, one utilizes an implicit approach where the score matching objective is approximated by a tractable alternative which may be easily computed through the derivatives of the approximating model.

The main contribution of this work is the identification of simple scenarios, even for the simple setting of the generalized Poisson process, where the score function optimization function of prior approaches fails as the regularity condition underlying them does not apply. The paper derives an expression showcasing the bias (Proposition 3.1) and is related to the evaluation of the estimated score function (for fixed number of occurrences) at the end points of the interval of interest ($t = 0$ and $t = T$). The paper then proposes an alternative where a weight function $h$ is used to augment the score function loss which ensures that these biasing terms evaluate to $0$. Furthermore, these functions may be chosen independently of the parameters of the underlying model. In their experiments, it is shown that the use of such functions still satisfies desirable properties such as the optima corresponding to the true values of the parameters while also allowing for a tractable score function formulation which avoids the bias incurred by prior approaches. This is a natural approach which suggests an avenue towards improving the performance of such models.

Overall, the problem considered in the paper and the solution they identify are interesting and likely relevant to the NeurIPS community. However, I do have concerns regarding the writing of the paper. For example, there are several assumptions on the weight function in Equation 9 but it is not clear which properties are used in the proof of Proposition 3.3. For instance, Line 400 features an explicit expression for $h_1$ and $h_N$ while the expression in Equation 9 features none of these. It is my understanding that the assumptions in Equation 9 ensure that the biasing terms equate to $0$ for these elements. Furthermore, I believe that under the assumptions in Equation 9, both terms appearing in the display between Lines 399 and 400 are $0$. Please clarify whether this is in fact the case. The same issues persist for the proofs of the results in Section 4. I would like clarification from the authors on these points before updating my review.

************************************************************************************************************************************************

I thank the authors for their response and have amended my review accordingly.

**Strengths:**

See main review

**Weaknesses:**

See main review

**Questions:**

See main review

**Limitations:**

See main review

---

> ### Author Rebuttal · Authors · 2024-08-07
>
> Thank you so much for reviewing our paper. We answer your questions below.
> > For example, there are several assumptions on the weight function in Equation 9 but it is not clear which properties are used in the proof of Proposition 3.3. For instance, Line 400 features an explicit expression for $h_1$ and $h_N$ while the expression in Equation 9 features none of these. Please clarify whether this is in fact the case. The same issues persist for the proofs of the results in Section 4.
>
> We greatly thank you for pointing this out. Your understanding is correct, and we apologize for the typos. The proofs for Propositions 3.3 and 4.3 are from our previous edition, where we specified one weight function instead of all weight functions that satisfy the condition. **We have proofread our work and ensured that the rest of our proofs remain correct.**
>
> For Propositions 3.3 and 4.3, the main parts remain correct. Only minor modifications are needed to utilize the assumptions stated in the main text, ensuring that the proofs hold for general weight functions. We have corrected the necessary parts as follows.
>
> From lines 398 to 404, the corrected proof is as follows:
>
> Using the first two equations in Eq. 9, we have
>
> $
> \int [p(t_1,\ldots, t_N)\frac{\partial \log p_{\theta}(t_1,\ldots, t_N)}{\partial t_{n}}h_n(\mathcal T)]\big\vert_{t_{n}=t_{n+1}}d\mathcal T_{-n}=0, \forall n\in [N]
> $
>
> $
> \int [p(t_1,\ldots, t_N)\frac{\partial \log p_{\theta}(t_1,\ldots, t_N)}{\partial t_n}h_n(\mathcal T)]\big\vert_{t_n=t_{n-1}}d\mathcal T_{-n}=0,\forall n\in [N].
> $
>
> Therefore, the first term on the right side of the second equation of Eq. 21 will disappear, and the second term equals $-\mathbb E_{p(\mathcal T)}[\sum_{n=1}^{N_T}\frac{\partial}{\partial t_n}\psi_{\theta}(t_n)h_n(\mathcal{T})+\psi_{\theta}(t_n)\frac{\partial}{\partial t_n}h_n(\mathcal T)]$.The existence of such an expectation is due to the last two equations in Eq. 9. Therefore, we complete the proof.
>
> We can see from the proof that, in Eq. 9, the first two equations ensure that the integration by parts trick does not produce an intractable term, and the last two equations are simply regularity conditions that ensure all terms are well-defined.
>
> Similarly, from line 431 to 434, the corrected proof is as follows:
>
> $
> \mathbb E_{p(\mathcal T)}[\sum_{n=1}^{N_T} \psi(t_n|\mathcal F_{t_{n-1}})\psi_\theta(t_n|\mathcal F_{t_{n-1}})h_n(\mathcal T)]
> $
>
> $
> =\sum_{n=1}^\infty \int p(t_1,\ldots t_n)\psi(t_n|\mathcal F_{t_{n-1}})\psi_\theta(t_n|\mathcal F_{t_{n-1}})h_n(\mathcal T)d\mathcal T_{1:n}
> $
>
> $
> =\sum_{n=1}^\infty \int p(\mathcal T_{:n-1})p(t_n|\mathcal F_{t_{n-1}})\psi_\theta(t_n|\mathcal F_{t_{n-1}})h_n(\mathcal T)\vert_{t_n=t_{n-1}}^{t_n=T}d\mathcal T_{:{n-1}}
> $
>
> $
> -\sum_{n=1}^\infty \int p(t_1,\ldots, t_n)[\frac{\partial \psi_\theta(t_n|\mathcal F_{t_{n-1}})}{\partial t_n}h_n(\mathcal T) + \psi_\theta(t_n|\mathcal F_{t_{n-1}})\frac{\partial h_n(\mathcal T)}{\partial t_n}]d\mathcal T_{1:n}.
> $
>
>
> Between the second and the third line above, we omit the steps used in the derivation of Proposition 4.1 to make it concise. For the term in the third line above, it will be eliminated using Eq. 15. For the term in the fourth line above, using Lemma B.1, we have:
>
> $
> -\sum_{n=1}^\infty \int p(t_1,\ldots, t_n)\left[\frac{\partial \psi_{\theta}(t_n|\mathcal F_{t_{n-1}})}{\partial t_n}h_n(\mathcal T) + \psi_{\theta}(t_n|\mathcal F_{t_{n-1}})\frac{\partial h_n(\mathcal T)}{\partial t_n}\right]d\mathcal T_{1:n}=\\
> -\mathbb E_{p(\mathcal T)}[\sum_{n=1}^{N_T}\frac{\partial \psi_{\theta}(t_n|\mathcal F_{t_{n-1}})}{\partial t_n}h_n(\mathcal T) + \psi_{\theta}(t_n|\mathcal F_{t_{n-1}})\frac{\partial h_n(\mathcal T)}{\partial t_n}].
> $
>
> The existence of the expectation is ensured by the last two terms of Eq. 15.
>
> We hope the corrected proof can address your concerns.

---

> ### Author Response · Authors · 2024-08-09
> **Thanks**
>
> Thank you very much for your thoughtful review and for taking the time to carefully consider our responses. We greatly appreciate your kind words and are pleased that you find our work to be well-presented, thorough, and novel.
>
> We will certaintly correct the proofs in the updated manuscript, as you suggested. Your feedback is valuable in improving the quality of our work, and we are grateful for your support.
>
> Thank you once again for your constructive feedback and for increasing your rating.

---

### Author Rebuttal · Authors · 2024-08-07

We express our sincere appreciation to all reviewers for their time, effort, and insightful feedback. We are encouraged by their recognition of the significance of our work in identifying the incompleteness of the original score matching for point processes, proposing the weighted score matching method, studying its convergence rate, proposing a near-optimal weight function, and deploying the method on deep point process models.

In the following, we have provided specific and direct responses to each of the primary concerns raised by the reviewers, supplemented with additional experiments that substantiate our arguments. We aim to ensure that we address all the concerns and offer clarity and reassurance where needed. Should any additional questions arise, we invite reviewers to engage in further discussion. Once again, we express our gratitude for your time and dedication in reviewing our work.

---

### Comment · Area_Chair_UHVE · 2024-08-07

Dear Reviewers,

Now that the rebuttal period has ended, please take a moment to review the authors' responses to your initial reviews. Your engagement is crucial for:

    Ensuring a fair evaluation process
    Improving the quality of final reviews
    Fostering meaningful scientific discourse

If you have any questions or need clarification, please post follow-up comments.

Your continued dedication to this process is greatly appreciated.

Best regards,

AC

---

### Decision · Program_Chairs · 2024-09-25

**Decision:**

Accept (poster)

**Comment:**

This paper addresses an important issue in applying score matching techniques to point process models, specifically Poisson and Hawkes processes. The authors identify a fundamental problem with existing score matching approaches for these models due to bounded support, and propose a novel weighted score matching solution to address this limitation.

All reviewers agree that the paper makes a valuable contribution by pointing out this critical issue and providing a principled solution. The theoretical analysis demonstrating the consistency of the proposed estimator and the empirical evaluations showing its effectiveness are viewed positively. The paper is generally well-written and the presentation is clear.

Some key strengths highlighted by reviewers include:
- Identifying an important limitation of existing score matching approaches for point processes
- Proposing a theoretically-grounded weighted score matching solution
- Thorough empirical evaluation demonstrating the effectiveness of the proposed method
- Clear presentation of the theory and assumptions

The main concerns raised relate to:
- Lack of comparison to some relevant baselines like denoising score matching
- Questions about the optimality of the proposed weighting function
- Some notational issues and need for clarification in parts

The authors have provided detailed responses addressing most of these concerns, including additional experimental results comparing different weighting functions. This has increased reviewer confidence in the work.

Overall, the reviewers recommend accepting this paper, contingent on the authors making a thorough revision incorporating the feedback and additional results from the discussion phase. Specifically, the authors should:

1. Include comparisons to denoising score matching approaches as discussed in the rebuttal
2. Add the additional experimental results on different weighting functions to the main paper
3. Expand the discussion on the choice and optimality of the weighting function
4. Clarify the notation, especially around equation 1 as mentioned by Reviewer X8YN
5. Consider adding MSE of event time prediction as an evaluation metric
6. Discuss recent related work on score matching for point processes

With these revisions, this paper promises to make a solid contribution to the field. The authors should carefully incorporate all the feedback received during the review process to strengthen the final version.